# Depth-discriminative Metric Learning for Monocular 3D Object Detection

**Wonhyeok Choi**[*]      **Mingyu Shin**[*]      **Sunghoon Im**[†]

DGIST, Daegu, Korea

{smu06117, alsrb4446, sunghoonim}@dgist.ac.kr

## Abstract

Monocular 3D object detection poses a significant challenge due to the lack of depth information in RGB images. Many existing methods strive to enhance the object depth estimation performance by allocating additional parameters for object depth estimation, utilizing extra modules or data. In contrast, we introduce a novel metric learning scheme that encourages the model to extract depth-discriminative features regardless of the visual attributes without increasing inference time and model size. Our method employs the distance-preserving function to organize the feature space manifold in relation to ground-truth object depth. The proposed $(K, B, \epsilon)$-quasi-isometric loss leverages predetermined pairwise distance restriction as guidance for adjusting the distance among object descriptors without disrupting the non-linearity of the natural feature manifold. Moreover, we introduce an auxiliary head for object-wise depth estimation, which enhances depth quality while maintaining the inference time. The broad applicability of our method is demonstrated through experiments that show improvements in overall performance when integrated into various baselines. The results show that our method consistently improves the performance of various baselines by 25.27% and 4.54% on average across KITTI and Waymo, respectively.

## 1 Introduction

Monocular 3D object detections [20, 22, 23, 27] has gained prominence as a cost-effective and easily deployable solution, playing a critical role in autonomous driving and robotic navigation systems. Typically, the frameworks consist of a feature extractor and lightweight multi-head architectures that predict the projected centers, depths, bounding box sizes, and heading directions of multiple objects from a single RGB image [18, 20, 22, 23, 26, 27, 41]. Among these tasks, object depth estimation, inferring the distance from a monocular camera to the center of an object, is the most challenging sub-task due to depth ambiguity. Previous work [23] has highlighted this issue in ablation studies, revealing that substituting object depth predictions with ground-truth (GT) values significantly improves overall performance, while replacing other sub-tasks, such as heading direction and 3D size, does not notably enhance performance. Furthermore, several prior works [3, 10, 18, 22, 27, 28, 37, 41] have attempted to improve object depth estimation quality by adding additional modules or introducing new formulations. However, despite the subsequent increase in inference time and model size, the improvement in performance remains limited.

These observations suggest that monocular 3D object detection heavily relies on object depth quality; nevertheless, conventional methods yield unsatisfactory results due to the extraction of less-discriminative features for object depth inference. One reason is that the extracted feature involves visual attributes, such as the color, size, and heading direction of objects, resulting in the object

---

[*]Equal Contribution

[†]Corresponding Author

37th Conference on Neural Information Processing Systems (NeurIPS 2023).

depth head receiving feature with limited depth discernment. To improve object depth performance, the network should be capable of extracting purely depth-discriminative features that involve essential geometric information, irrespective of the visual attribute. A feasible method for extracting the depth-discriminative features involves using deep metric learning schemes such as contrastive learning [6, 7, 39] or representation learning [5, 34].

However, most existing deep metric learning schemes [6, 7, 16, 39] rely on aggressive two-view augmentation (*i.e.* Affine transform) to train the distance or similarity metric. This aggressive data augmentation method is hardly leveraged in current monocular 3D object detection frameworks due to the violation of geometric constraints, where horizontal flip and color distortion are the only two methods used in this field for a long time [19]. One alternative way is to learn a regression-aware representation by contrasting samples against each other based on their target distance using GT labels [39]. However, forcibly arranging the manifold of the feature space using this depth distance metric may negatively impact the performance of the other tasks, because the feature extractor for monocular 3D object detection inevitably produces complex shared representations across multiple sub-tasks. Note that the experiment of negative impact is conducted in Sec. 5.2.

To address these issues, we propose a $(K, B, \epsilon)$-quasi-isometric loss, a new metric learning approach that encourages the network to extract depth-discriminative features using object depth labels. Inspired by several manifold learning schemes [29, 33], we locally preserve the neighborhood distances to maintain the natural non-linear manifold of feature space in order to mitigate the negative transfer effect on other sub-tasks. Our approach utilizes the quasi-isometric properties providing a relaxed condition for the distance metric between the depth and feature metric spaces. This enables the model to arrange the feature space with respect to the object depth labels while maintaining the performance of other sub-tasks. Moreover, we introduce an auxiliary head for object-wise depth estimation to further improve object depth estimation. The head component is removed after the training process, ensuring that the inference time remains unaffected and does not experience an increase. Experimental results indicate that the proposed method consistently outperforms state-of-the-art baselines across a variety of 3D object detection datasets. Our method has been shown to be compatible with various monocular 3D object detection frameworks [15, 20, 22, 23, 27], demonstrating its broad applicability without compromising inference time or increasing model size. The effectiveness of each proposed module is further underscored through comprehensive ablation studies.

Our contributions can be summarized as follows:

- We propose a simple yet effective metric learning scheme that preserves the geodesic distance of depth information to feature space.

- We present an auxiliary head for object-wise depth estimation, which enhances the depth quality without impacting the inference time, maintaining efficient performance.

- Our method significantly enhances the performance of various monocular 3D object detection methods without increasing inference time and model size.

## 2 Related work

**Monocular 3D object detection.** Monocular 3D object detection can be broadly categorized into two types. The first type [3, 18, 20, 22, 23, 25, 37, 41] predicts the localization of objects of interest around an ego vehicle using only RGB images, annotations, and camera calibrations. Most of them are based on CenterNet [43]. They are divided into several sub-tasks, with a primary focus on estimating accurate object depth. The works [22] and [41] propose the formulations of object depth estimation. MonoCon [20] leverages the abundant contexts in conventional 3D bounding box annotations to integrate various auxiliary tasks. The second category leverages additional data, such as pre-trained models [8, 10, 15, 27, 28, 35] or CAD models [24, 38], to compensate for the lack of 3D information in monocular images. Some prior works utilize pre-trained depth estimators to address the lack of depth information, either by converting the monocular setups to LiDAR/Stereo environments [8, 35] or by allocating more parameters to the depth estimation task [10, 15, 40]. Alternatively, certain methods employ completed depth maps from pre-trained depth completion models [27] or pre-trained LiDAR-based detectors [8, 14] to supervise their models. Notably, DID-M3D [27] divides object depth into visual and attribute depth using completed depth maps to address the ambiguity of the object depth.

**Manifold geometry preservation.**   Deep learning networks, consisting of continuous and differentiable (precisely, almost everywhere differentiable) layers, ensure smooth mapping of networks $f : \mathbb{R}^m \rightarrow \mathbb{R}^n$. This guarantees that neighborhoods in $\mathbb{R}^m$ will be mapped into neighborhoods in the embedding space $\mathbb{R}^n$ with some amount of "stretching" and vice versa. Manifold learning has been studied to preserve most or all of the essential information by minimizing these stretches. Many works [1, 2, 17, 29, 42] aim to create meaningful representations when embedding high-dimensional data into lower dimensions. The traditional linear embedding algorithms commonly use the matrix decomposition to preserve the original variance[1] or pairwise distance [17]. More recent algorithms [2, 29, 33, 42] construct the neighborhood graphs by preprocessing the dataset using $\epsilon$-ball or $k$-nearest neighbors for each point, embedding the data while preserving the neighbor samples' distance. In particular, Isomap [33] estimates the shortest path in the neighborhood graph between every pair of data points and then employs the Euclidean Multidimensional Scaling (MDS) algorithm [17] to embed the points in $d$ dimensions with minimum distance distortion all at once. These non-linear embedding algorithms can account for the non-linear nature of the manifold by preserving the distance between neighborhood samples.

**Metric learning.**   Metric learning is one of the machine learning techniques that aims to learn an effective distance metric between data points by using training data. To achieve this, methods such as [6, 7, 16] attempt to minimize the distance between samples from the same class while maximizing the distance between samples from different classes. These methods demonstrate improved performance for self-supervised learning by learning in an end-to-end fashion, including two-view augmentation or depending on class labels. These deep metric learning methods typically target classification tasks because the positive/negative pairs are defined as belonging to the same or different classes. More recent works [31, 36, 39] extensively apply metric learning in the context of regression. These works either directly use the target distance to contrast samples against each other [36, 39] or leverage representation learning in a semi-supervised learning scenario [31].

## 3   Method

Our final goal is to boost the object depth performance by extracting the depth-discriminative features without increasing inference time. Similar to common deep metric learning schemes [6, 7, 39], we propose a loss term that preserves a meaningful low-dimensional data structure in the feature space.

### 3.1   Preliminary

**Metric space.**   A metric space is a mathematical concept that characterizes a set of points and a function that measures the distance between any two points in the set. Formally, a metric space can be defined as a pair $(M, d)$, where $M$ represents a set and $d$ is a distance function on $M$. The distance function $d$ must satisfies the following axioms [13] for any three points $x, y, z \in M$:

1. Non-negativity: $d(x, y) \geq 0$ and $d(x, y) = 0$ if and only if $x = y$.
2. Symmetry: $d(x, y) = d(y, x)$ for all $x, y \in M$.
3. Triangle inequality: $d(x, y) \leq d(x, z) + d(z, y)$ for all $x, y, z \in M$.

**Quasi-isometry.**   A quasi-isometry is a function between two metric spaces that preserves distances up to a constant factor, even though it may locally distort angles and distances. Let $\mathcal{Q}$ be a function from one metric space $(M_1, d_1)$ to another metric space $(M_2, d_2)$. $\mathcal{Q}$ is considered a quasi-isometry from $(M_1, d_1)$ to $(M_2, d_2)$ if there exist constants $K \geq 1$, $B \geq 0$, and $\epsilon \geq 0$ such that both of the following properties hold:

1. $\forall x_1, x_2 \in M_1 \; : \; \frac{1}{K} \cdot d_1(x_1, x_2) - B \leq d_2(\mathcal{Q}(x_1), \mathcal{Q}(x_2)) \leq K \cdot d_1(x_1, x_2) + B.$
2. $\forall z \in M_2 \; : \; \exists x \in M_1 \; s.t. \; d_2(z, \mathcal{Q}(x)) \leq \epsilon.$

Quasi-isometry does not necessarily require continuity [4]. This property is advantageous because other distance-preserving transformations may not possess it, and most datasets are finite. Therefore, we use these conditions as constraints to ensure that the feature space retains the geometrical information of the depth metric space.

## 3.2 Problem Definition

The task of monocular 3D object detection aims to predict both the object class $c$ and the 3D bounding box $\mathbf{b}$ for multiple objects within an image $\mathbf{I}$. Recently, CenterNet [43] has become the best practice for monocular 3D object detection, with subsequent works [18, 20, 22, 23, 27, 41] adopting its pipeline. These methods decompose the bounding box $\mathbf{b} = [X, Y, Z, h, w, l, \gamma]$ estimation problem into separate estimations of the coarse projected 3D center $(u, v)$, the depth of object center $z$, the center offset $(\delta u, \delta v)$, the 3D size dimensions $(h, w, l)$, and heading direction $\gamma$ (*i.e.*, the yaw angle). The 3D object center $[X, Y, Z]$ is computed by back-projecting the projected center with the corresponding depth, given the intrinsic matrix of the camera $\mathbf{K}$, as follows:

$$
\begin{aligned}
u_c = u + \delta u, \; v_c = v + \delta v, \\
[X, Y, Z]^T = \mathbf{K}^{-1}[u_c \cdot z, v_c \cdot z, z]^T.
\end{aligned}
\tag{1}
$$

The networks consist of feature extractor $\mathcal{F}_\theta$ and task-specific heads $\mathcal{G}_{\phi^t}$ that produce the feature maps $\mathbf{h}$ and the per-pixel output maps $\tilde{\mathbf{o}}_t$, where $t \in \mathbf{T} = \{t_c, t_u, t_v, t_{\delta u}, t_{\delta v}, t_z, t_h, t_w, t_l, t_\gamma\}$, respectively, as follows:

$$
\begin{aligned}
\mathbf{h} = \mathcal{F}_\theta(\mathbf{I}), \; \mathcal{F}_\theta : \mathbb{R}^{3 \times H \times W} \to \mathbb{R}^{C \times H' \times W'}, \\
\tilde{\mathbf{o}}_t = \mathcal{G}_{\phi^t}(\mathbf{h}), \; \mathcal{G}_{\phi^t} : \mathbb{R}^{C \times H' \times W'} \to \mathbb{R}^{H' \times W'},
\end{aligned}
\tag{2}
$$

where $H, W$ represent the spatial resolution of the image, and $C, H', W'$ denote the channel and spatial resolution of the feature maps, which is downsampled from the image resolution. The final object-wise results $\mathbf{o_T}$ are extracted from the non-learnable function $\mathcal{H}$, given the intermediate per-pixel outputs $\tilde{\mathbf{o}}_{\mathbf{T} = \{t_c, t_u, t_v, t_{\delta u}, t_{\delta v}, t_z, t_h, t_w, t_l, t_\gamma\}}$, as follows:

$$
\mathbf{o_T} = \mathcal{H}(\tilde{\mathbf{o}}_\mathbf{T}), \; \mathcal{H} : \mathbb{R}^{|\mathbf{T}| \times H' \times W'} \to \mathbb{R}^{|\mathbf{T}| \times N}, \; N \geq 0,
\tag{3}
$$

where the number of detected objects $N$ is not fixed and can be zero if no object is detected.

## 3.3 Methodology

In this paper, our primary focus is on training the feature extractor $\mathcal{F}_\theta$ with the objective of extracting features that enhance the discriminability of object depth. At the same time, we aim to preserve the discriminability of other sub-tasks, such as the projected 3D center, bounding box size, and heading directions. To achieve this, we propose a metric learning method that encourages the network to extract depth-discriminative features by leveraging object depth labels.

Given $M$ training images $\mathbf{I}^i$, where $i \in I = \{1, ..., M\}$, and $N^i$ GT objects in the image $\mathbf{I}^i$, we extract object descriptors $\rho^{(i,j)}$, where $j \in J = \{1, ..., N^i\}$, from feature maps $\mathbf{h}^i$ using GT coarse projected 3D center $u^{(i,j)}, v^{(i,j)}$ and obtain the corresponding GT object depths $z^{(i,j)}$. We build a set of object descriptors $\mathbf{P} = \bigcup_{\{i \in I, \; j \in J\}} \rho^{(i,j)}$ and a set of corresponding object depth $\mathbf{Z} = \bigcup_{\{i \in I, \; j \in J\}} z^{(i,j)}$. We then define two metric spaces $(\mathbf{Z}, d_1)$ and $(\mathbf{P}, d_2)$, where $d_1$ and $d_2$ represent the Minkowski distance in Euclidean space (*i.e.*, L1 distance). The finite sets $\mathbf{Z}$ and $\mathbf{P}$ consist of $L$ elements, corresponding to objects in the dataset ($|\mathbf{Z}| = |\mathbf{P}| = L = \sum_{i \in I} N^i$). Consequently, we can establish a one-to-one function $\mathcal{Q}$ between these two metric spaces as follow:

$$
\mathcal{Q}(z^l) = \rho^l, \; \mathcal{Q} : \mathbf{Z} \to \mathbf{P}, \; l \in \{1, 2, \ldots, L\}.
\tag{4}
$$

Our goal is for the function $\mathcal{Q}$ to enforce a quasi-isometric between $\mathbf{Z}$-space and $\mathbf{P}$-space by using the properties of quasi-isometry in Sec. 3.1. This can encourage the network to extract depth-discriminative features in $\mathbf{P}$-space by utilizing object depth labels in $\mathbf{Z}$-space. However, enforcing a quasi-isometric between the low-dimensional $\mathbf{Z}$-space and the high-dimensional $\mathbf{P}$-space can damage the non-linearity of the natural manifold, potentially causing the negative transfer to other sub-tasks. Therefore, we adopt the local distance-preserving condition of the non-linear embedding methods such as Isomap [33] and LLE [29]. The revised version of the quasi-isometry condition is as follows:

(i) $\mathbf{U}_{\acute{z}} = \{z \in \mathbf{Z} | z \in \mathcal{B}_{\acute{z}, \epsilon}\}$,

(ii) $\forall z_1 \in \mathbf{Z}$, s.t. $\forall z_2 \in \mathbf{U}_{z_1} \; : \; \frac{1}{K} \cdot d_1(z_1, z_2) - B \leq d_2(\mathcal{Q}(z_1), \mathcal{Q}(z_2)) \leq K \cdot d_1(z_1, z_2) + B$,

(iii) $\forall \rho \in \mathbf{P} \; : \; \exists z \in \mathbf{Z}$, s.t. $d_2(\rho, \mathcal{Q}(z)) \leq \epsilon$,

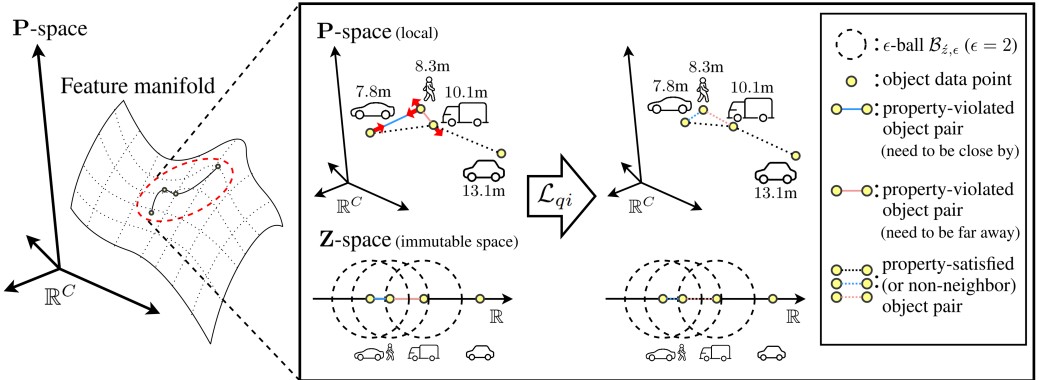

Figure 1: Illustration of our quasi-isometric loss $\mathcal{L}_{qi}$.

where $\mathcal{B}_{\acute{z},\epsilon}$ is a ball with radius $\epsilon$ centered around $\acute{z}$. By applying the quasi-isometric properties solely to neighboring samples within the $\epsilon$-ball, the non-linearity of the feature manifold is preserved, and the shortest curve distance between any two arbitrary samples on the manifold (*a.k.a.* geodesic) in **P**-space is maintained instead of the Minkowski distance. By ensuring these three conditions (i), (ii), (iii) met for a sufficiently small $K \geq 1$ and $B \geq 0$, we can encourage object features $\rho^l$ to be depth-discriminative using their corresponding object depth $z^l$. To achieve this, function $\mathcal{Q}$ must satisfy these conditions, which involves training the parameter $\theta$ of the feature extractor $\mathcal{F}$.

**(K, B, $\epsilon$)-Quasi-isometric loss.** We propose a quasi-isometric loss term that enforces the quasi-isometric between the depth and feature metric spaces while preserving the distance among the neighbor data points as illustrated in Fig. 1. In particular, the proposed loss term is designed to arrange **P**-space samples that do not satisfy these conditions. To ensure efficient training within the constraints of limited GPU memory, we use the samples within a mini-batch instead of using the entire input dataset as done in previous works [2, 29, 33]. We selectively choose the neighbor objects in a mini-batch with respect to their corresponding depth labels, where neighboring object depths are within $\mathbf{U}_z$.

Suppose that objects in mini-batch images $\mathbf{I}_b$ correspond to the object depths $\mathbf{Z}_b$ and object features $\mathbf{P}_b$ on **Z**-space and **P**-space, respectively. We can then identify the object feature pair sets $\mathbf{P}_b^+, \mathbf{P}_b^-$ that violate the revised property (ii) by transposing it as follow:

$$\forall (\rho_1, \rho_2) \in \mathbf{P}_b^+ \subset \mathbf{P}_b \times \mathbf{P}_b, \text{ s.t. } |z_1, z_2| \leq \epsilon : d_2(\rho_1, \rho_2) \not\leq K \cdot d_1(z_1, z_2) + B, \quad (5a)$$

$$\forall (\rho_1, \rho_2) \in \mathbf{P}_b^- \subset \mathbf{P}_b \times \mathbf{P}_b, \text{ s.t. } |z_1, z_2| \leq \epsilon : \frac{1}{K} \cdot d_1(z_1, z_2) - B \not\leq d_2(\rho_1, \rho_2), \quad (5b)$$

where $(z_1, z_2)$ is the corresponding depth pair of the object features $(\rho_1, \rho_2)$. For an object feature pair with distance $d_2(\rho_1, \rho_2)$ in **P**-space, the distance is larger for $\mathbf{P}_b^+$ in Eq. 5a or smaller for $\mathbf{P}_b^-$ in Eq. 5b than the corresponding depth pair distance $d_1(z_1, z_2)$ with a factor of $K$ and the additive constant $B$. To ensure these property-violating object feature pairs $\mathbf{P}_b^+, \mathbf{P}_b^-$ satisfy the property (ii), we propose the $(K, B, \epsilon)$-quasi-isometry loss, which modifies the Normalized Temperature-Scaled Cross-Entropy loss (NT-Xent loss) from [6] as follows:

$$\mathcal{L}_{qi} = -\frac{1}{|\mathbf{P}_b^+|} \sum_{(\rho_i, \rho_j) \in \mathbf{P}_b^+} \log \frac{\mathrm{S}^+(\rho_i, \rho_j)}{\mathrm{S}^+(\rho_i, \rho_j) + \sum_{(\rho_k, \rho_l) \in \mathbf{P}_b^-} \mathrm{S}^-(\rho_k, \rho_l)},$$

$$\mathrm{S}^+(\rho_i, \rho_j) = \exp(-(\|\rho_i, \rho_j\|_p - K|z_i, z_j| - B)/\tau), \quad (6)$$

$$\mathrm{S}^-(\rho_i, \rho_j) = \exp(-(\frac{1}{K}|z_i, z_j| - B - \|\rho_i, \rho_j\|_p)/\tau),$$

---
**Algorithm 1** $(K, B, \epsilon)$-Quasi-isometric loss

---
**Input:** $\mathbf{h}, (u_i, v_i), z_i$, where $i \in \{1, 2, ..., n\}$, $n$ is the number of objects in a batch.
  1: $\mathbf{h}_{avg} \leftarrow \text{avg\_pool\_5x5}(\mathbf{h})$ # feature map avg. pool
  2: initialize $\mathbf{P} = []$
  3: **for** $\forall$ i **do**
  4:     $\mathbf{P}[i] = \rho_i = \mathbf{h}_{avg}[:, u_i, v_i]$
  5: **end for**
  6: $\mathbf{Z} \leftarrow \{z_1, z_2, \ldots, z_n\}$ that correspond to $\mathbf{P}$.
  7: $\mathcal{M}_{\mathbf{P}} \leftarrow \begin{pmatrix} <\rho_1,\rho_1> & <\rho_1,\rho_2> & \cdots & <\rho_1,\rho_n> \\ \vdots & \vdots & \ddots & \vdots \\ <\rho_n,\rho_1> & <\rho_n,\rho_2> & \cdots & <\rho_n,\rho_n> \end{pmatrix}$,
                                        # where $< a, b >$ is the Minkowski metric $\|a, b\|_p$.
  8: $\mathcal{M}_{\mathbf{Z}} \leftarrow \begin{pmatrix} |z_1,z_1| & |z_1,z_2| & \cdots & |z_1,z_n| \\ \vdots & \vdots & \ddots & \vdots \\ |z_n,z_1| & |z_n,z_2| & \cdots & |z_n,z_n| \end{pmatrix}$
  9: $\mathcal{M}^+ \leftarrow \mathcal{M}_{\mathbf{P}} - K\mathcal{M}_{\mathbf{Z}} - B$ # To find property-violated object features. (Eq. 5a)
  10: $\mathcal{M}^- \leftarrow \frac{1}{K}\mathcal{M}_{\mathbf{Z}} - \mathcal{M}_{\mathbf{P}} - B$ # To find property-violated object features. (Eq. 5b)
  11: $(\mathcal{M}^+)_{ij} \leftarrow 0$, where $|z_i, z_j| > \epsilon$ or $i \geq j$ or $(\mathcal{M}^+)_{ij} < 0$.
  12: $(\mathcal{M}^-)_{ij} \leftarrow 0$, where $|z_i, z_j| > \epsilon$ or $i \geq j$ or $(\mathcal{M}^-)_{ij} < 0$.
  13: $\text{ancs}^+ \leftarrow \exp(-\mathcal{M}^+/\tau)$ # $\tau$ is temperature term.
  14: $\text{ancs}^- \leftarrow \sum(\exp(-\mathcal{M}^-/\tau))$
  15: $\mathcal{L}_{qi} = \text{mean}(-\log(\text{ancs}^+/(\text{ancs}^+ + \text{ancs}^- + \delta)))$ # $\delta = 1e - 12$: div. assert

---

where $\tau$ is the temperature term, and $\|\cdot, \cdot\|_p$ is $p$-norm, and $K \geq 1, B \geq 0$ denote the pre-defined hyperparameters that determine the hardness of quasi-isometric property. This loss aims to make the property-violated object features comply with the quasi-isometric property by assuming $\mathbf{P}_b^{+/-}$ as positive/negative anchors, respectively. The similarity metrics $\text{S}^{+/-}$ are always positive values in $(0, 1]$ that imply a negative distance gap between the distance metric on $\mathbf{Z}$-space and $\mathbf{P}$-space. These adjustments are necessary for proper alignment between the two spaces, noting that $\mathbf{P}_b$ is derived from $\mathcal{F}_\theta(\mathbf{I}_b)$.

**Object-wise depth map loss.** According to Eq. 1, the estimated object depth $\hat{z}$ and coarse projected 3D center $(\hat{u}, \hat{v})$ jointly determine the location of the object 3D center $(\hat{X}, \hat{Y}, \hat{Z})$. However, many existing methods [20, 21, 23, 27] train the depth loss on the local region exactly on the GT center $(u, v)$, which can lead to significant errors in object depth estimation even with slightly inaccurate object center $(\hat{u}, \hat{v})$ during inference (*i.e.*, center shifting in the image plane by a few pixels) [23].

To mitigate this issue, we train our network with an additional auxiliary head for object-wise depth estimation using the structure of [20]. Rather than solely providing depth supervision to an object center $(u, v)$, the auxiliary head is trained with depth supervision over the entire bounding box of the object. We create the foreground object-wise depth map $\mathfrak{D} \in \mathbb{R}^{H' \times W'}$ following the policy of [40]. Then, we define the object-wise depth loss by adopting the Laplacian aleatoric uncertainty loss [9], the same as the object depth estimation task, only for the foreground regions:

$$\mathcal{L}_{obj} = \frac{1}{|\mathfrak{D}|} \sum_{z^p \in \mathfrak{D}} \frac{\sqrt{2}}{\hat{\sigma}^p} |z^p - \hat{z}^p| + \log(\hat{\sigma}^p), \tag{7}$$

where $z^p, \hat{z}^p$ and $\hat{\sigma}^p$ are the corresponding GT object depth, prediction, and uncertainty of each pixel, respectively. Note that the additional task head is removed after training, so the proposed method does not increase the inference time.

**Total loss.** The total loss $\mathcal{L}_{total}$ combines the loss used in 3D object detection baselines $\mathcal{L}_{baseline}$, the quasi-isometric loss $\mathcal{L}_{qi}$, and the object-wise depth map loss $\mathcal{L}_{obj}$ as follows:

$$\mathcal{L}_{total} = \mathcal{L}_{baseline} + \lambda_{qi} \cdot \mathcal{L}_{qi} + \lambda_{obj} \cdot \mathcal{L}_{obj}, \tag{8}$$

where $\lambda_{qi}$ and $\lambda_{obj}$ represent the balancing weights for the quasi-isometric loss and the object-wise depth map loss, respectively. These values are set to 0.5 and 1. This total loss is applied to each baseline in our experiments, which we denote as "[Baseline] + Ours", as discussed further in Sec. 5.

Table 1: Evaluation results on KITTI *validation* set. (Blue/Red (positive/negative): relative performance from baseline.)

| Extra data | Method | Car, $AP_{3D|R40}$ ↑ | | | Car, $AP_{BEV|R40}$ ↑ | | |
| --- | --- | --- | --- | --- | --- | --- | --- |
| | | Easy | Moderate | Hard | Easy | Moderate | Hard |
| LiDAR | DID-M3D [27] | 23.93 | 16.22 | 13.98 | 32.65 | 23.15 | 19.48 |
| | DID-M3D + Ours | 24.77 (+3.5%) | 17.12 (+5.5%) | 14.30 (+2.3%) | 34.32 (+5.1%) | 23.45 (+1.3%) | 20.64 (+6.0%) |
| None | MonoDLE [23] | 17.32 | 14.35 | 12.22 | 24.62 | 20.25 | 17.75 |
| | MonoDLE + Ours | 21.31 (+23.0%) | 16.53 (+15.2%) | 13.93 (+14.0%) | 29.34 (+19.2%) | 22.27 (+10.0%) | 19.20 (+8.2%) |
| | GUP-Net [22] | 22.10 | 16.17 | 14.18 | 30.67 | 22.79 | 19.64 |
| | GUP-Net + Ours | 24.21 (+9.5%) | 17.82 (+10.2%) | 15.01 (+5.9%) | 32.38 (+5.6%) | 23.61 (+3.6%) | 21.20 (+7.9%) |
| | MonoCon [20] | 23.03 | 17.84 | 15.37 | 32.97 | 24.13 | 20.90 |
| | MonoCon + Ours | 27.90 (+21.1%) | 19.43 (+8.9%) | 16.92 (+10.3%) | 36.15 (+9.8%) | 26.15 (+8.6%) | 22.56 (+7.9%) |

Additionally, to simplify the implementation of our proposed $(K, B, \epsilon)$-quasi-isometric loss, we detail the loss function in Alg.1.

# 4 Experiments

**Datasets.** The KITTI dataset [12] consists of 7,481 training images and 7,518 test images for official KITTI 3D object detection evaluation and contains three categories: *Car*, *Pedestrian*, and *Cyclist*. For additional experiments, we follow [23], which splits the training images into 3,712 and 3,769 images for training and validation sets, respectively. The Waymo dataset [32] is a recently released dataset comprising 798 training sequences and 202 validation sequences, with four categories: *Vehicles*, *Pedestrians*, *Cyclists*, and *Signs*. We use the split reported in [28], including 52,386 training and 39,848 validation images, to evaluate performance on the Waymo dataset.

**Evaluation Metrics.** We adhere to the protocol reported in [12] and [28] for KITTI and Waymo datasets, respectively. The KITTI 3D object detection performance is evaluated by the average precision of 3D bounding boxes ($AP_{3D|R40}$) with IoU thresholds of 0.7 for *Car* and 0.5 for *Pedestrian* and *Cyclist*. The evaluation is split into three levels of difficulty: Easy, Moderate, and Hard, based on the 2D bounding box height, occlusion level, and truncation. In contrast, the evaluation metrics of the Waymo dataset are based on 3D IoU with mean average precision (3D mAP) and mean average precision weighted by heading (3D mAPH). Each object is divided into one or two levels and evaluated at three distances: [0, 30), [30, 50), and [50, ∞) meters.

**Implementation details.** First, we incorporate our method into four different baselines, including CenterNet-based frameworks [20, 22, 23, 27] for the KITTI dataset. For the Waymo dataset, we choose [20], as it performs best on the KITTI dataset among our baselines. We primarily follow the experimental setup details (*i.e.*, epochs, optimizer) of each paper [20, 22, 23, 27] for a fair comparison. We denote the plugged versions of baselines as "[Baseline] + Ours". As an exception, we set all batch sizes to 16 for the KITTI and Waymo benchmarks. We retrain all baselines on the KITTI/Waymo *validation* set with "*Car*" and "*Vehicle*" classes, respectively. For the $(K, B, \epsilon)$-quasi-isometric loss, we use $K = 1.5, B = 0.5, \epsilon = 10.0, d_1(\cdot, \cdot) = |\cdot, \cdot|, d_2(\cdot, \cdot) = \|\cdot, \cdot\|_2$ for all experiments. We provide more detailed experimental setups for each baseline in the supplementary materials.

# 5 Experimental Results

## 5.1 Evaluation Results on KITTI and Waymo Datasets

**KITTI dataset.** We demonstrate the effectiveness of our method on both the KITTI validation and test datasets. As shown in Tab. 1, our method can be widely applied to various baselines and significantly enhances performance by a considerable margin. Particularly, the improvement is more pronounced in models that do not utilize extra data, surpassing models that do (13.1% vs. 3.8%). We surmise that the model trained with additional LiDAR data learns more discriminative features regarding depths than the method without extra data. Notably, the performance of "MonoCon + Ours" exceeds that of models trained with additional LiDAR data. This result illustrates that our proposed metric learning method empowers MonoCon to extract depth-discriminative features without the need for extra data. Similar trends are observable in the evaluation with the KITTI test datasets, as shown in Tab. 2. The experiments demonstrate that our proposed method enhances depth discrimination not

Table 2: Evaluation results on KITTI *test* set.

| Extra data | Method | Car, $AP_{3D|R40}$ ↑ | | | Ped., $AP_{3D|R40}$ ↑ | | | Cyc., $AP_{3D|R40}$ ↑ | | |
|---|---|---|---|---|---|---|---|---|---|---|
| | | Easy | Mod. | Hard | Easy | Mod. | Hard | Easy | Mod. | Hard |
| LiDAR | DID-M3D [27] | 24.40 | 16.29 | 13.75 | - | - | - | - | - | - |
| | DID-M3D + Ours | 27.04 | 16.42 | 13.37 | 14.41 | 9.05 | 8.05 | 4.86 | 3.11 | 2.97 |
| | *Improvement* | (+10.8%) | (+0.8%) | (−2.8%) | - | - | - | - | - | - |
| None | MonoDLE [23] | 17.23 | 12.26 | 10.29 | 9.64 | 6.55 | 5.44 | 4.59 | 2.66 | 2.45 |
| | MonoDLE + Ours | 22.11 | 15.30 | 12.72 | 11.75 | 7.80 | 6.29 | 6.02 | 4.12 | 3.42 |
| | *Improvement* | (+28.3%) | (+24.8%) | (+23.6%) | (+21.9%) | (+19.1%) | (+15.6%) | (+31.2%) | (+54.9%) | (+39.6%) |
| | GUPNet [22] | 20.11 | 14.20 | 11.77 | 14.72 | 9.53 | 7.87 | 4.18 | 2.56 | 2.09 |
| | GUPNet + Ours | 23.19 | 15.78 | 13.02 | 14.23 | 9.03 | 8.06 | 5.68 | 3.61 | 3.13 |
| | *Improvement* | (+15.3%) | (+11.1%) | (+10.6%) | (−3.3%) | (−5.2%) | (+2.4%) | (+35.9%) | (+41.0%) | (+49.8%) |
| | MonoCon [20] | 22.50 | 16.46 | 13.95 | 13.10 | 8.41 | 6.94 | 2.80 | 1.92 | 1.55 |
| | MonoCon + Ours | 23.31 | 16.36 | 13.73 | 14.90 | 10.28 | 8.70 | 5.38 | 2.89 | 2.83 |
| | *Improvement* | (+3.6%) | (−0.6%) | (−1.6%) | (+13.7%) | (+22.2%) | (+25.4%) | (+92.1%) | (+50.5%) | (+82.6%) |

Table 3: Evaluation results on Waymo *validation* set.

| Difficulty | Method | Vehicle, $AP_{3D}$ ↑ | | | | Vehicle, $APH_{3D}$ ↑ | | | |
|---|---|---|---|---|---|---|---|---|---|
| | | Overall | 0-30 m | 30-50m | 50m-∞ | Overall | 0-30 m | 30-50 m | 50 m-∞ |
| LEVEL_1 | MonoCon [20] | 2.30 | 6.66 | 0.67 | 0.02 | 2.29 | 6.62 | 0.66 | 0.02 |
| | MonoCon + Ours | 2.50 | 7.62 | 0.72 | 0.02 | 2.48 | 7.57 | 0.72 | 0.02 |
| (IOU = 0.7) | *Improvement* | (+8.7%) | (+14.4%) | (+7.5%) | (+0.0%) | (+8.3%) | (+14.4%) | (+9.1%) | (+0.0%) |
| LEVEL_2 | MonoCon | 2.16 | 6.64 | 0.64 | 0.02 | 2.15 | 6.59 | 0.64 | 0.02 |
| | MonoCon + Ours | 2.34 | 7.59 | 0.70 | 0.02 | 2.33 | 7.54 | 0.69 | 0.02 |
| (IOU = 0.7) | *Improvement* | (+8.3%) | (+14.3%) | (+9.4%) | (+0.0%) | (+8.4%) | (+14.4%) | (+7.8%) | (+0.0%) |
| LEVEL_1 | MonoCon | 10.07 | 27.47 | 3.84 | 0.16 | 9.99 | 27.26 | 3.81 | 0.16 |
| | MonoCon + Ours | 10.14 | 28.51 | 3.99 | 0.17 | 10.06 | 28.28 | 3.96 | 0.17 |
| (IOU = 0.5) | *Improvement* | (+0.7%) | (+3.8%) | (+3.9%) | (+6.3%) | (+0.7%) | (+3.7%) | (+3.9%) | (+6.3%) |
| LEVEL_2 | MonoCon | 9.44 | 27.37 | 3.71 | 0.14 | 9.37 | 27.17 | 3.68 | 0.14 |
| | MonoCon + Ours | 9.50 | 28.40 | 3.85 | 0.15 | 9.43 | 28.18 | 3.82 | 0.15 |
| (IOU = 0.5) | *Improvement* | (+0.6%) | (+3.8%) | (+3.8%) | (+7.1%) | (+0.6%) | (+3.7%) | (+3.8%) | (+7.1%) |

only for *Car* but also for *Pedestrian* and *Cyclist*. The average performance increases for these object categories are 10.3%, 12.4%, and 53.1%, respectively.

**Waymo dataset.** We extend the evaluations to the Waymo dataset to further demonstrate the generality of our method, as shown in Tab. 3. The results indicate that our proposed method consistently enhances performance by an average of 4.6% and 4.5% in terms of mAP and mAPH metrics across all levels, IoU thresholds, and distance ranges. Significantly, the proposed loss boosts the depth discrimination not only within specific depth ranges but across all depth ranges. This underscores the effectiveness of our method not just on the KITTI dataset, but on the Waymo dataset as well.

## 5.2 Additional Experiments

**Ablation studies.** To demonstrate the effectiveness of our proposed method, we conduct the ablation studies of two components of our method: $(K, B, \epsilon)$-quasi-isometric loss $\mathcal{L}_{qi}$ and object-wise depth map loss $\mathcal{L}_{obj}$. We employ DID-M3D [27] and MonoCon [20] as baselines of a method with and without extra data. The results in Tab. 4 indicate that any method incorporating either the quasi-isometric or object-wise depth losses sees a performance improvement. Notably, the $(K, B, \epsilon)$-quasi-isometric loss contributes to a larger performance gain than the object-wise depth loss (+2.3% vs. +1.9% for DID-M3D and +8.7% vs. +5.0% for MonoCon). The models trained with both of the proposed losses significantly surpass the performance of the baselines.

**Comparison of our method with SupCR.** To demonstrate the advantage of our quasi-isometric loss over existing metric learning schemes, we compare the task performance of our method with that of SupCR (Supervised Contrastive Regression) [39]. SupCR is the first regression-aware representation learning method that effectively applies metric learning to regression tasks using GT labels. Similar to our quasi-isometric loss $\mathcal{L}_{qi}$, SupCR selectively chooses the relative negative pair object features based on positive pair distance. In Tab. 5, we report the $AP_{3D|R40}$ for *Car, Moderate* and errors between the GT and prediction of four key tasks that determine the location of the 3D bounding box: $t \in z, (h, w, l), \gamma, c$, independently. The errors include the absolute difference $E_z, E_{dim}$ for $z, (h, w, l)$, respectively, the mean angular distance $\Delta\gamma$ [11] for $\gamma$, and the accuracy $Acc._c$ for object

Table 4: Ablation studies on KITTI *validation* set.

| Components | | Baseline | Car, $AP_{3D\|R40}$ ↑ | | | Overall |
| $\mathcal{L}_{qi}$ | $\mathcal{L}_{obj}$ | | Easy | Moderate | Hard | |
|---|---|---|---|---|---|---|
| | | DID-M3D [27] | 23.93 | 16.22 | 13.98 | 0.0% |
| √ | | | 24.61 (+2.8%) | 16.57 (+2.2%) | 14.23 (+1.8%) | +2.3% |
| | √ | | 24.53 (+2.5%) | 16.84 (+3.8%) | 13.87 (−0.8%) | +1.8% |
| √ | √ | | 24.77 (+3.5%) | 17.12 (+5.5%) | 14.30 (+2.3%) | +3.8% |
| | | MonoCon [20] | 23.03 | 17.84 | 15.37 | 0.0% |
| √ | | | 27.00 (+17.2%) | 18.68 (+4.7%) | 16.02 (+4.2%) | +8.7% |
| | √ | | 24.90 (+8.1%) | 18.65 (+4.5%) | 15.73 (+2.3%) | +5.0% |
| √ | √ | | 27.90 (+21.1%) | 19.43 (+8.9%) | 16.92 (+10.3%) | +13.5% |

Table 5: Comparison of our method with SupCR on KITTI *validation* set.

| Baseline | $AP_{3D\|R40}$ ↑ | $E_z$ (m) ↓ | $E_{dim}$ (m) ↓ | $\Delta\gamma$ (rad) ↓ | $Acc._c$ (%) ↑ |
|---|---|---|---|---|---|
| MonoCon [20] | 17.84 | 0.019 | 0.025 | $\pi/371.79$ | 93.15 |
| MonoCon + SupCR | 17.55 (−1.6%) | 0.014 (−26.3%) | 0.031 (+24.0%) | $\pi/312.78$ (+18.9%) | 91.11 (−2.2%) |
| MonoCon + $\mathcal{L}_{qi}$ | 18.68 (+4.7%) | 0.011 (−42.1%) | 0.024 (−4.0%) | $\pi/368.11$ (+1.0%) | 93.07 (−0.1%) |

classification $c$. The results illustrate that our method enhances 3D object detection performance, whereas SupCR diminishes it. Interestingly, both methods reduce depth errors $E_z$ as they train the encoder to extract more depth-discriminative features. However, SupCR significantly increases the errors in bounding box size and angle estimation $E_{dim}, \Delta\gamma$ by approximately 24.0% and 18.9%, respectively. It demonstrates that SupCR, which forcibly arranges the feature manifold, can negatively impact the performance of other tasks due to the complex shared representations across multiple sub-tasks.

**Scalability of our quasi-isometric loss with the anchor-based method and BEV paradigm.** In Tab. 1-4, we note that our method improves performance when applied to CenterNet-based baselines (anchor-free). We further demonstrate the effectiveness of the proposed metric learning incorporated into anchor-based methods and the bird-eye-view (BEV) paradigm: MonoDTR [15] and ImVoxelNet [30]. The object-wise depth map loss is not applied to these baselines, since it already uses an auxiliary depth loss or inherent nature of the BEV paradigm.

As shown in Tab. 6, although the performance enhancement compared to anchor-free methods is not as significant, our quasi-isometric loss still shows consistent improvements across all metrics for each baseline. These results suggest that the loss has broad applicability to the tasks that extract object features on image spatial coordinates given object depth labels.

**Model performance with respect to the ratio of property-violated objects.** The proposed quasi-isometric loss is designed to arrange the feature space in accordance with object depth labels, aiming to meet quasi-isometric properties. To demonstrate the correlation between 3D object detection performance and the ratio of features that satisfy the quasi-isometric condition, we conduct an additional experiment. In Fig. 2, we plot the ratio of object pairs violating the properties (where the Ratio $= (|\mathbf{P}^+|+|\mathbf{P}^-|)/|\mathbf{P}|$, $|\mathbf{P}| = \binom{L}{2}$, and $L$ is the total number of objects in the dataset.) and model performance in relation to various loss scales $\lambda_{qi}$. For this exper-

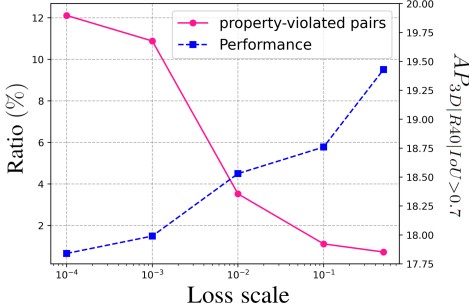

Figure 2: Loss scale (log-scaled) with respect to property-violated pairs ratio and performance.

iment, we use the baseline model [20] trained with different loss scales $[10^{-5}, 10^{-3}, 10^{-2}, 10^{-1}, 0.5]$. We measure the $AP_{3D}$ with IoU thresholds of 0.7 for *Car, Moderate* on the KITTI validation dataset. The results indicate that an increase in the loss scale corresponds with a decrease in the ratio of object pairs violating the quasi-isometric properties, leading to an improvement in model performance. This

Table 6: Evaluation results of anchor-based and bird-eye-view paradigms on KITTI *validation* set.

| Method | Car, $AP_{3D\|R40}$ ↑ | | | Overall |
| | Easy | Moderate | Hard | |
| --- | --- | --- | --- | --- |
| MonoDTR [15] | 24.57 | 18.45 | 15.37 | - |
| MonoDTR + Ours | 26.17 (+6.5%) | 19.07 (+3.4%) | 15.93 (+3.6%) | +4.5% |
| ImVoxelNet [30] | 24.54 | 17.80 | 15.87 | - |
| ImVoxelNet + Ours | 26.14 (+8.0%) | 18.20 (+7.2%) | 15.81 (+3.8%) | +6.3% |

Table 7: Performance of KITTI *validation* with respect to $K, B, \epsilon$.

| Components | | | Car, $AP_{3D\|R40}$ ↑ | | | Overall |
| K | B | $\epsilon$ | Easy | Moderate | Hard | |
| --- | --- | --- | --- | --- | --- | --- |
| MonoCon (w/o $\mathcal{L}_{obj}$) | | | 23.03 | 17.84 | 15.37 | +0.0% |
| 1.0 | 0.5 | 10.0 | 24.67 (+7.1%) | 18.09 (+1.4%) | 15.34 (−0.2%) | +2.8% |
| 1.5 | 0.5 | 10.0 | **27.00** (+17.2%) | 18.68 (+4.7%) | **16.02** (+4.2%) | **+8.7%** |
| 2.0 | 0.5 | 10.0 | 23.34 (+1.3%) | 17.44 (−2.2%) | 14.80 (−3.7%) | -1.5% |
| 1.5 | 0.0 | 10.0 | 24.54 (+6.6%) | 18.01 (+1.0%) | 15.37 (+0.0%) | +2.5% |
| 1.5 | 1.0 | 10.0 | 25.49 (+10.7%) | **18.70** (+4.8%) | 15.82 (+2.9%) | +6.1% |
| 1.5 | 5.0 | 10.0 | 23.70 (+2.9%) | 18.00 (+0.9%) | 15.12 (+-1.6%) | +0.9% |
| 1.5 | 0.5 | 1.0 | 24.71 (+7.3%) | 17.84 (+0.0%) | 15.43 (+0.4%) | +2.6% |
| 1.5 | 0.5 | 5.0 | 24.99 (+8.5%) | 18.23 (+2.2%) | 15.80 (+2.8%) | +4.5% |
| 1.5 | 0.5 | $\infty$ | 22.85 (−0.8%) | 16.91 (−5.2%) | 14.14 (−8.0%) | -4.7% |

suggests a potential relationship between the reduction in the ratio of property-violated object pairs and the enhancement of model performance.

**Influence of hyperparameters** $(K, B, \epsilon)$ **in quasi-isometric loss.** The choice of the tunable hyperparameter $\epsilon$ is crucial. An excessively small $\epsilon$ would sample a few objects within a mini-batch, making representation learning infeasible. On the other hand, quasi-isometric properties with too large an $\epsilon$ would harm the non-linearity of the feature manifold in **P**-space. To investigate the influence of hyperparameters on our quasi-isometric loss, we evaluate the performance of "Monocon + Ours" using KITTI [12] validation set for various hyperparameters $(K, B, \epsilon)$, as presented in Tab. 7. Although most hyperparameter combinations improved performance, there are notable exceptions, especially when $K = 2.0$ and $\epsilon = \infty$. We observe a decline in performance with excessively large values of $K$ or $\epsilon$, which define the criteria for property violation and the non-linearity of the feature manifold.

## 6   Conclusion

In this paper, we address the challenge of monocular 3D object detection in RGB images by proposing a novel metric learning scheme. Our method, which does not rely on extra parameters, modules, or data, concentrates on extracting depth-discriminative features without increasing the inference time or model size. By employing a distance-preserving function and the $(K, B, \epsilon)$-quasi-isometric loss, we successfully arrange the feature space manifold in accordance with ground-truth object depth, while preserving the non-linearity of the natural feature manifold. Furthermore, by introducing an auxiliary head for object-wise depth estimation, we improve object depth quality without increasing inference time. Our experimental results on the KITTI and Waymo datasets illustrate consistent performance enhancements across different baselines, highlighting the effectiveness of our proposed method. As a potential avenue for future work, our method could feasibly be extended to multi-camera 3D object detection scenarios and other regression tasks that involve multiple sub-tasks.

## Acknowledgments and Disclosure of Funding

This work was supported by the National Research Foundation of Korea (NRF) grant funded by the Korea government (MSIT) (No. RS-2023-00210908).

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
