## A  General Notations

In Tab. 1, we provide a comprehensive summary of the general notations used throughout the paper to illustrate our framework and clarify the formulation of our methodology.

Table 1: General notations

| Datasets | |
|---|---|
| Image | $\mathbf{I} \; : \; \mathbb{R}^{3 \times H \times W}$ |
| $i$-th image | $\mathbf{I}^i \; : \; \mathbb{R}^{3 \times H \times W}$ |
| Image set in mini-batch | $\mathbf{I}_b \; : \; \mathbb{R}^{B \times 3 \times H \times W}$ |
| Camera intrinsic matrix | $\mathbf{K}$ |
| Coarse projected 3D center | $(u, v)$ |
| Object depth | $z$ |
| Center offset | $(\delta u, \delta v)$ |
| 3D size dimensions | $(h, w, l)$ |
| Heading direction | $\gamma$ |
| Number of training images | $M$ |
| Number of objects in $\mathbf{I}^i$ | $N^i$ |
| Number of objects in entire images | $L$ |

| Sets, States | |
|---|---|
| Task set | $\mathbf{T} = \{t_c, t_u, t_v, t_{\delta u}, t_{\delta v}, t_z, t_h, t_w, t_l, t_\gamma\} \ni t$ |
| Image index set | $I = \{1, 2, \ldots, M\}$ |
| object index set in image $\mathbf{I}^i$ | $J = \{1, 2, \ldots, N^i\}$ |
| Feature maps (hidden state) | $\mathbf{h} \; : \; \mathbb{R}^{C \times H' \times W'}$ |
| Per-pixel output maps | $\tilde{\mathbf{o}}_{\mathbf{T}} \; : \; \mathbb{R}^{|\mathbf{T}| \times H' \times W'}$ |
| Object-wise output (prediction) | $\mathbf{o}_{\mathbf{T}} \; : \; \mathbb{R}^{|\mathbf{T}| \times N^i}$ |
| Object descriptor | $\rho \; : \; \mathbb{R}^{C \times 1 \times 1}$ |
| Distance metric | $d(\cdot, \cdot)$ |
| Object descriptor metric space | $(\mathbf{P}, d) \equiv \mathbf{P}\text{-space}$ |
| Object depth metric space | $(\mathbf{Z}, d) \equiv \mathbf{Z}\text{-space}$ |

| Network components, Function | |
|---|---|
| Feature extractor | $\mathcal{F}_\theta(\cdot) \; : \; \mathbb{R}^{3 \times H \times W} \rightarrow \mathbb{R}^{C \times H' \times W'}$ |
| Task-specific lightweight head | $\mathcal{G}_{\phi^t}(\cdot) \; : \; \mathbb{R}^{C \times H' \times W'} \rightarrow \mathbb{R}^{H' \times W'}$ |
| Extract function | $\mathcal{H}(\cdot) \; : \; \mathbb{R}^{|\mathbf{T}| \times H' \times W'} \rightarrow \mathbb{R}^{|\mathbf{T}| \times N^i}$ |
| Quasi-isometry | $\mathcal{Q}(\cdot) \; : \; \mathbb{R} \rightarrow \mathbb{R}^C$ |

## B  Theoretical analysis

This section provides our theoretical analysis of the proposed $(K, B, \epsilon)$-quasi-isometric loss term, which leverages the quasi-isometric properties between two metric spaces. This analysis clarifies how our method alleviates the bottleneck task (*i.e.*, object depth estimation) through mathematical theorems and empirical observations. Ideally, we aim for the quasi-isometric loss, with real finite data points on $\mathbf{P}$-space, to function similarly to its continuous counterpart $\mathcal{M}$ (*a.k.a.* True manifold). Essentially, as the number of object feature data points $|\mathbf{P}|$ approach to $\infty$, we intend for the object feature set $\mathbf{P}$ originating from backbone $\mathcal{F}_\theta$ to continue to fulfill the revised quasi-isometric properties, regardless of the particular sample, in a probabilistic sense.

Fig. 2 in our main manuscript presents empirical demonstrations that, when applying the proposed quasi-isometric loss with adequate weight term $\lambda_{qi}$, the ratio of property-violated object features converges to zero. Hence, we suppose that the network trained with quasi-isometric loss consistently produces the object features that adhere to the revised quasi-isometric properties. Moreover, we observe that hyperparameters $B$ and $\epsilon$ associated with our quasi-isometric loss term should be small enough as the number of data points in the $\mathbf{P}$-space incrementally approaches infinity. In this section,

we define the **pseudo-geodesic** to approximate the geodesic on the true manifold. We further establish that the feature space, which satisfies the local quasi-isometric properties, also adheres to the global quasi-isometric properties. Notably, in this context, the distance metric of the **P**-space is substituted by the pseudo-geodesic.

## B.1  Pseudo-geodesic

The search for the true geodesic on **P**-space manifold is impeded by the discontinuous finite data points in the set **P**, rendering the true manifold unobservable. Therefore, we establish the pseudo-geodesic $\hat{\mathcal{G}}(\rho_s, \rho_t)$ to approximate the length of the shortest curve between two data points $\rho_s, \rho_t$ on a feature manifold. This pseudo-geodesic is defined by partitioning the interval $[z_s, z_t] \subset \mathbb{R}$, denoted by $\mathcal{P}$, as follows:

$$
\hat{\mathcal{G}}(\rho_s, \rho_t) = \sum_{i=1}^{n} \|\rho_{i-1}, \rho_i\|_p,
$$
$$
\mathcal{P}_{\mathbf{P}} = (\rho_0, \rho_1, \ldots, \rho_n), \ \mathcal{P}_{\mathbf{Z}} = (z_0, z_1, \ldots, z_n), \tag{1}
$$
$$
\text{s.t. } z_s = z_0 < z_1 < z_2 < \cdots < z_n = z_t,
$$
$$
\max\{|z_{i-1} - z_i| : i = 1, 2, \ldots, n\} \le \delta < \epsilon,
$$

where $\mathcal{P}_{\mathbf{P}}$ is the sequence of data points in $|\mathbf{P}|$-space that corresponds to partition $\mathcal{P}_{\mathbf{Z}}$, $n$ is the number of the pseudo-geodesic $\mathcal{P}_{\mathbf{P}}$ curve segments, and $\delta$ is a sufficiently small scalar that ensures each curve segment length in $\mathcal{P}_{\mathbf{P}}$ is smaller than $\epsilon$. Note that $\mathcal{P}_{\mathbf{Z}}$ is a subset of $\{z \in \mathbf{Z}|z_s \le z \le z_t\}$, because pseudo-geodesic should represent the shortest path between $\rho_s$ and $\rho_t$. Given that the mesh of $\mathcal{P}_{\mathbf{Z}}$ is less than $\delta$, the pseudo-geodesic should remain close to the true manifold so as $\delta$ converges to zero, thereby approximating the length of the true geodesic. The defined pseudo-geodesic metric for any arbitrary object feature pair $(\rho_1, \rho_2)$ in **P**-space satisfies the properties of a distance metric.

**Definition B.1** (Quasi-isometric Properties), Let $\mathcal{Q}$ represent a function that maps one metric space $(M_1, d_1)$ to another metric space $(M_2, d_2)$. $\mathcal{Q}$ is termed a quasi-isometry from $(M_1, d_1)$ to $(M_2, d_2)$ if there exist constants $K \ge 1$, $B \ge 0$, and $\epsilon \ge 0$ such that the following two properties are satisfied:

(i) $\forall x_1, x_2 \in M_1 \ : \ \frac{1}{K} d_1(x_1, x_2) - B \le d_2(\mathcal{Q}(x_1), \mathcal{Q}(x_2)) \le K d_1(x_1, x_2) + B$.

(ii) $\forall z \in M_2 \ : \ \exists x \in M_1 \ s.t. \ d_2(z, \mathcal{Q}(x)) \le \epsilon$.

**Definition B.2** (Local Quasi-isometric Properties), Local quasi-isometry refers to a function whereby any two neighboring points $(x_1, x_2)$ in the domain set $M_1$ comply with the **Definition B.1**, with $x_2 \in \mathcal{B}_{x_1, \epsilon}$. Let $\mathcal{Q}$ be a function from one metric space $(M_1, d_1)$ to another metric space $(M_2, d_2)$. $\mathcal{Q}$ is considered a quasi-isometry from $(M_1, d_1)$ to $(M_2, d_2)$ if there exist constants $K \ge 1$, $B \ge 0$, and $\epsilon \ge 0$ that satisfy the following conditions:

(i) $\mathbf{U}_{\acute{x}} = \{x \in M_1 | x \in \mathcal{B}_{\acute{x}, \epsilon}\}$, where $\acute{x} \in M_1$.

(ii) $\forall x_1 \in M_1$, s.t. $\forall x_2 \in \mathbf{U}_{x_1} : \frac{1}{K} d_1(x_1, x_2) - B \le d_2(\mathcal{Q}(x_1), \mathcal{Q}(x_2)) \le K d_1(x_1, x_2) + B$,

(iii) $\forall z \in M_2 \ : \ \exists x \in M_1$, s.t. $d_2(z, \mathcal{Q}(x)) \le \epsilon$.

**Theorem.** *Given that $B = B'/|\mathbf{P}|$ and $B' \ge 0$, the two metric spaces $(\mathbf{Z}, |\cdot, \cdot|)$ and $(\mathbf{P}, \hat{\mathcal{G}}(\cdot, \cdot))$ are quasi-isometric.*

*Proof.* Let $(\mathbf{Z}, |\cdot, \cdot|)$ and $(\mathbf{P}, \|\cdot, \cdot\|_p)$ be two metric spaces. Assume that for all $(z_i, z_j) \in \mathbf{Z} \times \mathbf{Z}$ and $(\rho_i, \rho_j) \in \mathbf{P} \times \mathbf{P}$, the local quasi-isometric properties defined in **Definition B.2** are satisfied. We need to show that these pairs also satisfy the **Definition B.1** when the distance metric of **P**-space is the pseudo-geodesic $\hat{\mathcal{G}}$. We proceed as follows:

$$
\frac{1}{K}|z_{i-1} - z_i| - B \le \|\rho_{i-1}, \rho_i\|_p \le K|z_{i-1} - z_i| + B \qquad \text{(by **Definition B.2**)}
$$

Since this inequality holds for all $i$, summing over all $i$ from 1 to $|\mathcal{P}_{\mathbf{Z}}|$, we have

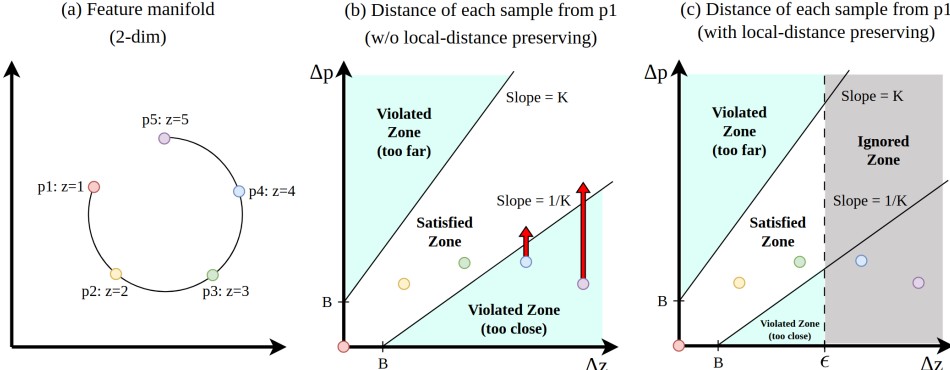

Figure 1: Example of non-linearity preservation by using local-constraint.

$$\sum_{i=1}^{|\mathcal{P}_\mathbf{z}|}\left(\frac{1}{K}|z_{i-1}-z_i|-B\right) \le \sum_{i=1}^{|\mathcal{P}_\mathbf{z}|}\|\rho_{i-1},\rho_i\|_p \le \sum_{i=1}^{|\mathcal{P}_\mathbf{z}|}(K|z_{i-1}-z_i|+B)$$

$$\implies \sum_{i=1}^{|\mathcal{P}_\mathbf{z}|}\left(\frac{1}{K}|z_{i-1}-z_i|-B\right) \le \hat{\mathcal{G}}(\rho_s,\rho_t) \le \sum_{i=1}^{|\mathcal{P}_\mathbf{z}|}(K|z_{i-1}-z_i|+B) \qquad \text{(by Eq. 1)}$$

$$\implies \frac{1}{K}|z_s-z_t|-|\mathcal{P}_\mathbf{z}|B \le \hat{\mathcal{G}}(\rho_s,\rho_t) \le K|z_s-z_t|+|\mathcal{P}_\mathbf{z}|B \qquad \text{(by Eq. 1)}$$

$$\implies \frac{1}{K}|z_s-z_t|-|\mathbf{P}|B \le \hat{\mathcal{G}}(\rho_s,\rho_t) \le K|z_s-z_t|+|\mathbf{P}|B \qquad (\because |\mathcal{P}_\mathbf{z}| \le \mathbf{P})$$

$$\implies \frac{1}{K}|z_s-z_t|-B' \le \hat{\mathcal{G}}(\rho_s,\rho_t) \le K|z_s-z_t|+B' \qquad (\because B=\frac{B'}{|\mathbf{P}|})$$

$$\implies (\mathbf{Z},|\cdot,\cdot|) \underset{q.i.}{\sim} (\mathbf{P},\hat{\mathcal{G}}(\cdot,\cdot))$$

$$\square$$

The distance metric defined as $\hat{\mathcal{G}}(\cdot,\cdot)$ satisfies all axioms of a distance function, and two metric spaces $(\mathbf{Z},|\cdot,\cdot|),(\mathbf{P},\hat{\mathcal{G}}(\cdot,\cdot))$ conform to the properties in **Definition B.1** with respect to $(K,B',\epsilon)$. This theorem implies that, by establishing an appropriate $B$ with respect to the number of objects in the entire dataset, the local quasi-isometric properties roughly preserve the pseudo-geodesic distance as opposed to the Minkowski distance on $\mathbf{P}$-space. This is analogous to stating that the pseudo-geodesic between two arbitrary points $\rho_s,\rho_t$ on $\mathbf{P}$-space is uniformly close to $|z_s-z_t|$.

### B.2 Non-linearity preservation

The proposed quasi-isometric loss benefits from the incorporation of a local distance-preserving condition. This ensures a structured arrangement of the feature manifold while maintaining its intricate overall shape. For instance, suppose that there is the feature space being modeled by a subset of the circle manifold, as depicted in Fig. 1-(a). From a depth perspective, this manifold represents a structured feature space since the distance of all object feature pairs along the geodesic corresponds closely with depth distance. However, without the local distance-preserving condition, as shown in Fig. 1-(b), the quasi-isometric loss might erroneously infer that features $p4$ and $p5$ violate property norms.

On the other hand, integrating the local distance-preserving condition denoted by $\epsilon$ (Fig. 1) refines the quasi-isometric loss to only consider neighbor samples. This approach enables a more nuanced arrangement of the feature manifold while preserving the overall shape and non-linearity of the original feature space.

In Tab. 2, we report $AP_{3D|R40}$ for *Car, Moderate* and errors between the GT and prediction of four key tasks that determine the location of the 3D bounding box: $t \in z, (h,w,l), \gamma, c$, independently.

Table 2: Performance trade-off between *"depth"* and *"other sub-tasks"* with respect to $\epsilon$.

| Method | Performance | Depth | Others | | |
|---|---|---|---|---|---|
| | $AP_{3D\|R40}$ ↑ | $E_z$ (m) ↓ | $E_{dim}$ (m) ↓ | $\Delta\gamma$ (rad) ↓ | $Acc._c$ (%) ↑ |
| MonoCon | 17.84 | 0.019 | 0.025 | $\pi/371.79$ | 93.15 |
| MonoCon + $\mathcal{L}_{\text{SupCR}}$ | 17.55 (−1.6%) | 0.014 (−26.3%) | 0.031 (+24.0%) | $\pi/312.78$ (+18.9%) | 91.11 (−2.2%) |
| MonoCon + $\mathcal{L}_{qi}$ ($\epsilon = 1$) | 17.84 (+0.0%) | 0.018 (−5.3%) | 0.025 (+0.0%) | $\pi/371.21$ (+0.2%) | 93.17 (+0.0%) |
| MonoCon + $\mathcal{L}_{qi}$ ($\epsilon = 5$) | 18.23 (+2.2%) | 0.013 (−31.6%) | 0.025 (+0.0%) | $\pi/\mathbf{372.10}$ (−0.1%) | **93.51** (+0.4%) |
| MonoCon + $\mathcal{L}_{qi}$ ($\epsilon = 10$) | **18.68** (+4.7%) | 0.011 (−42.1%) | **0.024** (−4.0%) | $\overline{\pi/368.11}$ (+1.0%) | 93.07 (−0.1%) |
| MonoCon + $\mathcal{L}_{qi}$ ($\epsilon = 20$) | 18.12 (+1.6%) | 0.011 (−42.1%) | 0.026 (+4.0%) | $\pi/366.19$ (+1.5%) | 92.00 (−1.2%) |
| MonoCon + $\mathcal{L}_{qi}$ ($\epsilon = \infty$) | 16.91 (−0.9%) | **0.010** (−47.4%) | 0.028 (+12.0%) | $\pi/367.92$ (+1.1%) | 91.83 (−1.4%) |

The empirical results illustrate that our method with excessively small or high $\epsilon$ would sample a few objects within a mini-batch, making representation learning infeasible, or harming the non-linearity of the feature manifold in $\mathbf{P}$-space, respectively.

## C  More detailed experimental setups

Table 3: Experimental setup of each baseline (horizontal flip: hf, random crop: rc, scaling: s, photometric distortion: pd, random shifting: rs).

| baseline | batch | epoch | image resolution | optimizer | augmentation type |
|---|---|---|---|---|---|
| DID-M3D [7] | 16 | 150 | 1280×384 | Adam [2] | hf, rc, s |
| MonoDLE [6] | 16 | 140 | 1280×384 | Adam | hf, rc, s |
| GUPNet [5] | 16 | 140 | 1280×384 | Adam | hf, rc, s |
| MonoCon [3] | 16 | 200 | 1248×384 | AdamW [4] | hf, pd, rs |
| MonoCon [3](Waymo [8]) | 16 | 50 | 768x512 | AdamW | hf, pd, rs |
| MonoDTR [1] | 16 | 120 | 1280×288 | Adam | hf |

As mentioned in the paper, our experimental settings are the same as the respective baselines except for the batch size. We elaborate on the specifics of the experimental setup in Tab. 3. For all baselines, we use DLA, DLAUp [9] as the backbone and neck, respectively. We impose our quasi-isometric loss and object-wise depth map loss using the output feature extracted from DLAUp. When computing the quasi-isometric loss, we first apply a 5x5 average pooling to the output feature prior to the extraction of the object descriptor. This extracted object descriptor is subsequently utilized to compute the loss. Regarding the object-wise depth map loss, we abide by the lightweight head structure adopted in each baseline and introduce an additional head. The output feature derived from DLAUp serves as the input, and the object-wise depth map is generated as the output. This depth map forms the basis for the computation of the loss.

## D  Qualitative results on KITTI dataset

We provide additional qualitative results using the MonoCon and "MonoCon + Ours" as discussed in Tab. 1 of the main manuscript, utilizing the KITTI *validation* set. In Fig. 2, we showcase the predictions of "MonoCon + Ours" in the image view on the left, while on the right, we present the Bird's Eye view displaying the predictions of Monocon, "MonoCon + Ours", and the GT. Generally, the models employing our method tend to align more closely with the GT.

## E  Full evaluation results on KITTI *test* set

Finally, we report the full evaluation results of four baseline models [3, 5–7] on KITTI *test* set in Tab. 4-6.

Table 4: Full evaluation results of *Car* class on KITTI ***test*** set.

| Extra data | Method | Car, $AP_{3D|R40}$ ↑ | | | Car, $AP_{BEV|R40}$ ↑ | | |
|---|---|---|---|---|---|---|---|
| | | Easy | Mod. | Hard | Easy | Mod. | Hard |
| LiDAR | DID-M3D [7] | 24.40 | 16.29 | 13.75 | 32.95 | 22.76 | 19.83 |
| | DID-M3D + Ours | 27.04 (+10.8%) | 16.42 (+0.8%) | 13.37 (−2.8%) | 34.77 (+5.5%) | 22.59 (−0.7%) | 19.15 (−3.4%) |
| None | MonoDLE [6] | 17.23 | 12.26 | 10.29 | 24.79 | 18.89 | 16.00 |
| | MonoDLE + Ours | 22.11 (+28.3%) | 15.30 (+24.8%) | 12.72 (+23.6%) | 30.28 (+22.1%) | 20.99 (+11.1%) | 17.73 (+10.8%) |
| | GUPNet [5] | 20.11 | 14.20 | 11.77 | 30.29 | 21.19 | 18.20 |
| | GUPNet + Ours | 23.19 (+15.3%) | 15.78 (+11.1%) | 13.02 (+10.6%) | 32.45 (+7.1%) | 22.31 (+5.3%) | 18.32 (+0.7%) |
| | MonoCon [3] | 22.50 | 16.46 | 13.95 | 31.12 | 22.10 | 19.00 |
| | MonoCon + Ours | 23.31 (+3.6%) | 16.36 (−0.6%) | 13.73 (−1.6%) | 32.37 (+4.0%) | 22.73 (+2.9%) | 19.81 (+4.3%) |

Table 5: Full evaluation results of *Ped.* class on KITTI ***test*** set.

| Extra data | Method | Ped, $AP_{3D|R40}$ ↑ | | | Ped, $AP_{BEV|R40}$ ↑ | | |
|---|---|---|---|---|---|---|---|
| | | Easy | Mod. | Hard | Easy | Mod. | Hard |
| LiDAR | DID-M3D [7] | - | - | - | - | - | - |
| | DID-M3D + Ours | 14.41 | 9.05 | 8.05 | 15.70 | 10.20 | 8.62 |
| None | MonoDLE [6] | 9.64 | 6.55 | 5.44 | 10.73 | 6.96 | 6.20 |
| | MonoDLE + Ours | 11.75 (+21.9%) | 7.80 (+19.1%) | 6.29 (+15.6%) | 12.85 (+19.8%) | 8.75 (+25.7%) | 7.31 (+17.9%) |
| | GUPNet [5] | 14.72 | 9.53 | 7.87 | - | - | - |
| | GUPNet + Ours | 14.23 (−3.3%) | 9.03 (−5.2%) | 8.06 (+2.4%) | 15.50 | 10.16 | 8.65 |
| | MonoCon [3] | 13.10 | 8.41 | 6.94 | - | - | - |
| | MonoCon + Ours | 14.90 (+13.7%) | 10.28 (+22.2%) | 8.70 (+25.4%) | 16.29 | 10.88 | 9.31 |

Table 6: Full evaluation results of *Cyc.* class on KITTI ***test*** set.

| Extra data | Method | Cyc, $AP_{3D|R40}$ ↑ | | | Cyc, $AP_{BEV|R40}$ ↑ | | |
|---|---|---|---|---|---|---|---|
| | | Easy | Mod. | Hard | Easy | Mod. | Hard |
| LiDAR | DID-M3D [7] | - | - | - | - | - | - |
| | DID-M3D + Ours | 4.86 | 3.11 | 2.97 | 5.94 | 4.02 | 3.55 |
| None | MonoDLE [6] | 4.59 | 2.66 | 2.45 | 5.34 | 3.28 | 2.83 |
| | MonoDLE + Ours | 6.02 (+31.2%) | 4.12 (+54.9%) | 3.42 (+39.6%) | 8.33 (+56.0%) | 5.64 (+72.0%) | 4.83 (+70.7%) |
| | GUPNet [5] | 4.18 | 2.56 | 2.09 | - | - | - |
| | GUPNet + Ours | 5.68 (+35.9%) | 3.61 (+41.0%) | 3.13 (+49.8%) | 6.47 | 3.85 | 3.82 |
| | MonoCon [3] | 2.80 | 1.92 | 1.55 | - | - | - |
| | MonoCon + Ours | 5.38 (+92.1%) | 2.89 (+50.5%) | 2.83 (+82.6%) | 7.07 | 4.06 | 3.85 |

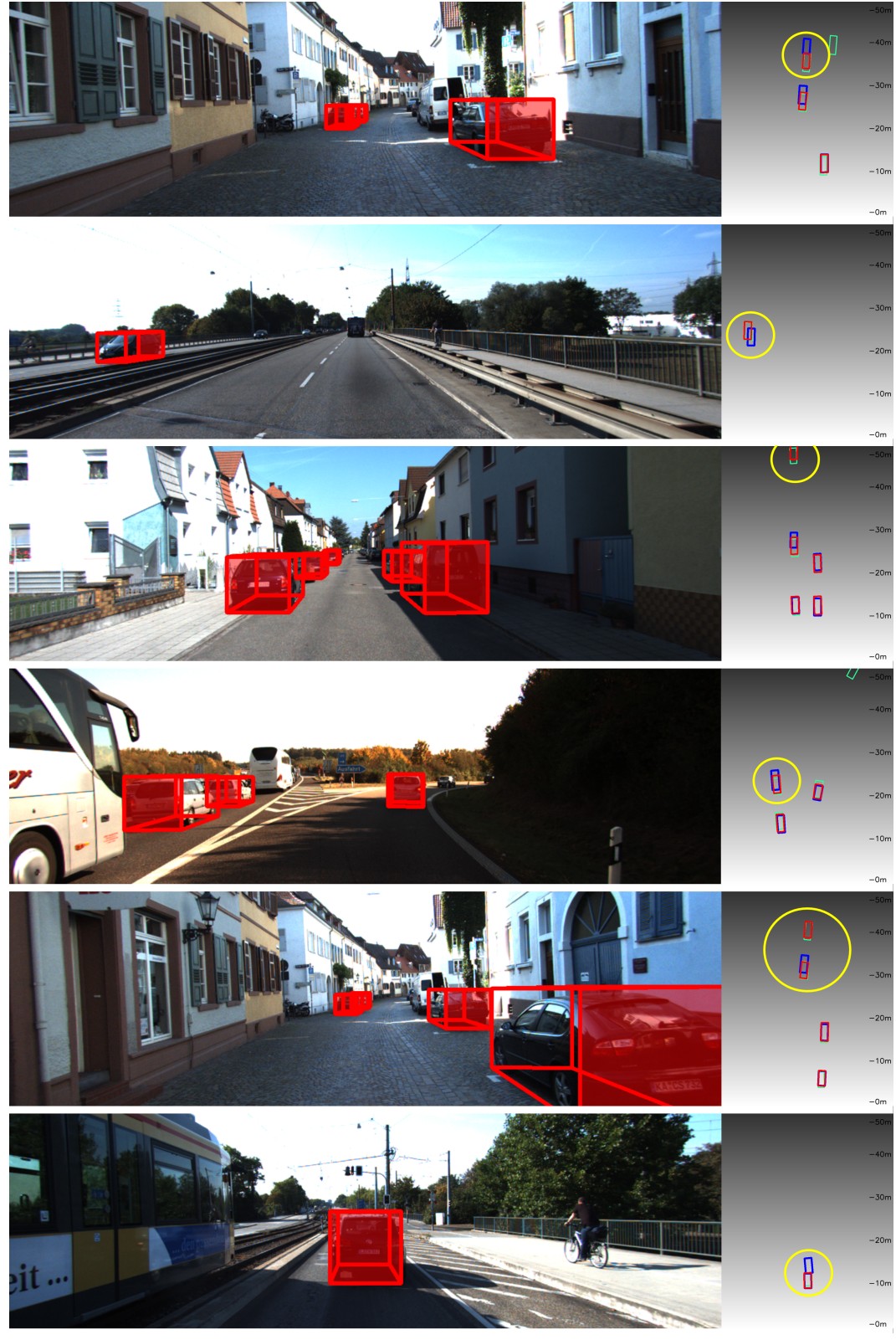

Figure 2: Comparison of qualitative results between MonoCon and MonoCon + ours. Yellow circles highlight accurately estimated parts compared to the MonoCon.
(GT: green, Prediction of MonoCon: blue, Prediction of MonoCon + Ours: red)