# OpenReview forum: "Depth-discriminative Metric Learning for Monocular 3D Object Detection"
_NeurIPS.cc/2023/Conference — NeurIPS 2023 poster_

### Official Review · Reviewer_eeNQ · 2023-07-01

**Soundness:** 3 good
**Presentation:** 4 excellent
**Contribution:** 3 good
**Rating:** 8
**Confidence:** 5

**Summary:**

This work focuses on the monocular 3D object detection task. As many works indicated, depth estimation is the bottleneck of this task, and the authors propose the apply metric learning to improve the accuracy of the depth estimation sub-task. The proposed metric-learning-based loss encourages the model to extract depth-discriminative features regardless of the visual attributes without increasing inference time and model size. Extensive experiments on the KITTI3D and Waymo Open datasets demonstrate the effectiveness of the proposed loss fuction.

**Strengths:**

1. I like the idea that applying metric learning to improve the accuracy of depth estimation. Theoretically, this method has a good generalization ability and is easy to be embedded in other baseline models.
2. Extensive experiments and good performance. The authors conduct lots of experiments on the KITTI3D and the large-scale Waymo Open datasets. Besides, they test their metric learning-based loss function on several baseline models, and the improvements across several settings demonstrate the effectiveness of their proposed model.
3. The main idea is easy to follow and this paper is well-organized. This paper has many mathematical details that scholars in this field may not be familiar with, but it still presents them clearly and systematically.



**Weaknesses:**

1. In lines 68-69, the authors claim their proposed work is 'the first approach that applies metric learning to monocular 3D object detection.'  In fact, there is another work [1] that discussed how to apply metric learning in monocular 3D object detection with a focus on dimension estimation. This claim should be modified or removed.

2. Although this work provides lots of experiments to show the effectiveness of their method, lots of them are based on the same foundation model, i.e. CenterNet (and one of them is based on the transformer-based pipeline). However, there are still some other popular detection pipelines such as the BEV paradigm. It will be better to validate the effectiveness of more pipelines.

3. This work conducts experiments on the large-scale Waymo dataset, which is great. However, I still want to see the performance on the nuScenes dataset, because the nuScenes benchmark is dominated by another detection pipeline and adopts different evaluation metrics. Evaluating on this dataset can further show the effectiveness and generalization ability of the proposed method.


Overall, this paper proposes a novel and effective metric-learning-based loss function. I tend to accept this work and I can further improve my rating if the authors can show the generalization ability of this work in more settings.

[1] Dimension Embeddings for Monocular 3D Object Detection, Zhang et al., CVPR'22

**Questions:**

See weaknesses.

**Limitations:**

See weaknesses.

---

> ### Author Rebuttal · Authors · 2023-08-09
>
> >**Q1.** In lines 68-69, the authors claim their proposed work is 'the first approach that applies metric learning to monocular 3D object detection.' In fact, there is another work that discussed how to apply metric learning in monocular 3D object detection with a focus on dimension estimation. This claim should be modified or removed.
>
> **A1.** We were not previously aware of the cited work. We appreciate your bringing it to our attention and will make the necessary revisions accordingly.
> ___
> >**Q2.** Although this work provides lots of experiments to show the effectiveness of their method, lots of them are based on the same foundation model, *i.e.* CenterNet (and one of them is based on the transformer-based pipeline). However, there are still some other popular detection pipelines such as the BEV paradigm. It will be better to validate the effectiveness of more pipelines.
>
> **A2.** We greatly appreciate your insightful feedback.
> Following your request, we conducted an additional experiment on ImVoxelNet [9] that adopts the BEV paradigm using the 3D voxel feature.
> This baseline reformulated the outdoor 3D object detection as 2D detection in the BEV plane. We thus extracted the object feature from the projected BEV feature prior to the head that consists of two parallel 2D convolution layers.
>
> Results, as shown in `R-Table 3`, show that while the performance gain isn't as pronounced as with CenterNet-based methods, the proposed $\cal{L_{qi}}$ consistently improves performance across all metrics, resulting in an overall performance boost of $+6.6$%.
> These findings, as highlighted in the main paper, underscores the broad utility of $\cal{L_{qi}}$ in 3D object detection, provided object depth labels are available.
> ___
> >**Q3.** This work conducts experiments on the large-scale Waymo dataset, which is great. However, I still want to see the performance on the nuScenes dataset, because the nuScenes benchmark is dominated by another detection pipeline and adopts different evaluation metrics. Evaluating this dataset can further show the effectiveness and generalization ability of the proposed method.
>
> **A3.** In response to your suggestion, we have now expanded our evaluation to include the nuScenes dataset [4], as presented in `R-Table 4`.
> The proposed method achieved a $+2.8$% improvement in MAE, and a notable $+12.3$% enhancement in Car AP, resulting in an overall performance boost of $+7.6$%.
> These consistent gains across all datasets underline the robustness and generalization ability.
> ___

---

> > ### Comment · Reviewer_eeNQ · 2023-08-11
> > **Final Rating**
> >
> > I appreciate the additional experiments which further confirm the effectiveness and generalization ability of the proposed method. Based on the solid theoretical analysis and good results, I'd like to change my score to 'strong accept'.

---

### Official Review · Reviewer_tr2v · 2023-07-06

**Soundness:** 3 good
**Presentation:** 3 good
**Contribution:** 3 good
**Rating:** 6
**Confidence:** 4

**Summary:**

This paper introduces a novel metric learning scheme for extracting more depth-discriminative features in the monocular 3D object detection task. A distance-preserving function is adopted to build the relation between feature space and the ground-truth object depth. The authors propose a quasi-isometric loss to adjust the distances among object descriptors. Furthermore, an auxiliary head for object-wise depth estimation is used during training, enhancing depth quality while maintaining the inference time. The experiments show that the proposed method can improve the performance significantly of various baselines on KITTI and Waymo datasets.

**Strengths:**

- This paper introduces a metric learning method to improve the discrimination of object descriptors according to their depth information. The scheme is simple yet effective, which maintains the geodesic distance of depth information in feature space. The quasi-isometry under the local distance-preserving condition mitigates the negative damage to the non-linearity of the natural manifold. It minimizes the influence on other tasks.

- The quasi-isometric loss is designed to arrange condition-violated samples. The samples far from the quasi-isometric distance are pulled closer. The samples too close in feature space are pushed away. The loss focuses on hard samples which avoids many unnecessary computations while ensuring the isometric objective.

- The proposed method is plug and play. It can easily integrate into various monocular 3D object detection baselines and brings significant improvement.

**Weaknesses:**

- I am curious about the efficiency of the proposed method during training. The calculations of relative distances between objects in both original and feature space would be enormous. Besides, the complexity of calculating the distance matrix grows with the square of the number of objects. Therefore, the proposed method may be not suitable for the scenario with a large number of objects during training. The paper does not discuss this problem.


- The object-wise depth prediction head is trained for estimating more accurate depth, avoiding significant errors that existed in the center depth prediction head. Why not directly use the prediction of object-wise depth head for a more accurate depth estimation during inference?

**Questions:**

See weaknesses.

**Limitations:**

The authors do not address the limitations of their work. This method may be limited by the number of objects.

---

> ### Author Rebuttal · Authors · 2023-08-09
>
> >**Q1.** I am curious about the efficiency of the proposed method during training. The calculations of relative distances between objects in both original and feature space would be enormous. Besides, the complexity of calculating the distance matrix grows with the square of the number of objects. Therefore, the proposed method may be not suitable for the scenario with a large number of objects during training. The paper does not discuss this problem.
>
> **A1.**
> As you pointed out, the time/memory complexity of calculating the distance matrix increases quadratically with the number of objects.
> Nevertheless, we enhanced the efficiency of our quasi-isometric method at the implementation level by loop unrolling as detailed in the algorithm table of supplementary materials.
> Instead of computing each distance pair-by-pair, we substituted these operations as matrix operations, which are GPU-friendly operations.
>
> Following your request, we empirically measure the training cost of our quasi-isometric loss.
> We compared the training time of one epoch between the 'MonoCon' [6] and 'MonoCon+Ours' using a single NVIDIA Titan RTX.
> We simulated the most challenging scenario by setting the maximum number of property-violated object pairs to 30 per image, given that MonoCon can infer up to 30 objects.
>
> Method|Avg. training time (sample/ms)|Single epoch time (s)
> ---|---|---
> MonoCon|$84.9$|$315$
> MonoCon+Ours (worst case)|$87.1~(+2.59$%$)$|$324~(+2.86$%$)$
>
> The results were as follows: 'MonoCon' required 315 seconds for one epoch, whereas 'MonoCon+Ours' took 324 seconds.
> This represents a mere $2.86$% increase in training time, which we believe has a negligible impact on overall computational efficiency.
> ___
> >**Q2.** The object-wise depth prediction head is trained for estimating more accurate depth, avoiding significant errors that existed in the center depth prediction head. Why not directly use the prediction of object-wise depth head for a more accurate depth estimation during inference?
>
> **A2**. When utilizing object-wise depth maps for depth prediction, issues such as occlusion arise.
> For instance, if the estimated center point is obscured by the 2D bounding box of a foreground car, the system might mistakenly extract the depth information of the foreground vehicle.
> This can lead to a decline in performance.
> Experimental results confirm this: 'MonoCon+Ours' achieved a car class mAP (mean average precision) of $19.43$ (moderate), whereas 'MonoCon+Ours using object-wise depth map for prediction depth' only managed a car class mAP of $15.17$ (moderate).
> ___

---

> > ### Comment · Reviewer_tr2v · 2023-08-14
> > **The reply addresses my concerns**
> >
> > Thanks for your reply. The feedback has addressed my concerns. After reading other reviews and the rebuttal materials, I lean to acccept this paper.

---

### Official Review · Reviewer_nNTt · 2023-07-07

**Soundness:** 3 good
**Presentation:** 3 good
**Contribution:** 3 good
**Rating:** 7
**Confidence:** 4

**Summary:**

One main challenge of monocular 3D object detection models is the lack of depth information from RGB images. The authors proposed a metric learning scheme to encourage the model to extract depth-discriminative features. Based on the presented theoretical results, the authors proposed a quasi-isometric loss and an object-wise depth map loss to supervise the model. Quantitative results on benchmark datasets supported the main arguments of the paper and sufficient ablation study experiments were conducted.

**Strengths:**

1. The motivation and problem setting are explained clearly. References were provided to demonstrate why depth is key to improve current monocular 3D methods and how depth-discriminative features would help.
2. The proposed quasi-isometric loss seems novel and effective. To prevent the depth-discriminative loss from damaging the non-linearity of the natural manifold, the authors adopted a distance-preserving condition. Despite extra hyper-parameters introduced, the proposed method is effective in general under various settings.
3. Quantitative results on two benchmark datasets using multiple baseline models demonstrated the effectiveness of the method. Results showed that the proposed depth-discriminative is an effective approach to assist monocular 3D methods.

**Weaknesses:**

1. The high-level idea of the proposed approach resembles previous contrastive learning approaches [1]. I could imagine adding depth-based feature contrastive losses to the baseline loss. Would that work? What would be the advantage of the proposed approach compared to contrastive losses?
2. The authors claimed that directly learning quasi-isometry would hurt sub-tasks. Despite such trade-offs being common in deep learning, I think this problem is a bit understudied. For instance, object features from objects that differ a lot in distance should be easily discriminated against. Would larger $K$ help with a larger $\epsilon$? Or is this design related to hard negative sampling?
3. Figure 2 seems interesting. However, I assume the pairs are also defined by the given $K, B, \epsilon$, which makes it less convincing.

References:
1. D. Neven et al. Towards End-to-End Lane Detection: an Instance Segmentation Approach.

**Questions:**

1. What is “Ours” in supplementary Table 3? Only $\mathcal{L}\_{qi}$ or both $\mathcal{L}\_{qi}$ and $\mathcal{L}\_{obj}$? I assume when ablating $K, B, \epsilon$, you should not involve $\mathcal{L}\_{obj}$ in the comparison?
2. Since there is an auxiliary head for depth estimation, is the estimated depth more accurate with the quasi-isometric loss? It would be good to compare the trade-off between depth estimation and “other sub-tasks” when $\epsilon$ changes.

**Limitations:**

1. The work can be improved by comparing with methods with similar ideas (see weakness 1).
2. It is a bit unclear how different feature spaces look like under different hyper-parameter settings. It would be good if more analysis are presented (see weakness 2).

---

> ### Author Rebuttal · Authors · 2023-08-09
>
> >**Q1.** The high-level idea of the proposed approach resembles previous contrastive learning approaches. I could imagine adding depth-based feature contrastive losses to the baseline loss. Would that work? What would be the advantage of the proposed approach compared to contrastive losses?
>
> **A1.**
> In our main paper, we excluded results from conventional metric learning methods like SimCLR [1] and SupCon [2] since they are primarily tailored for classification tasks.
> Instead, we chose to compare ours with SupCR [3], which is designed specifically for regression tasks.
> This is because the methods [1,2] are not directly applicable to object depth estimation as a regression task.
>
> To highlight the advantages of our approach over conventional contrastive losses, we modified two renowned contrastive learning techniques: SimCLR [1] and SupCon [2].
> SupCon is specifically tailored to enhance feature learning by harnessing the information derived from the GT depth (refer to `R-Table 1` for details).
>
> Due to space constraints, we kindly ask you to refer to **A2-2** of **R1 (Wa1k)** for a comprehensive analysis of the experiment.
> ___
> >**Q2.** The authors claimed that directly learning quasi-isometry would hurt sub-tasks. Despite such trade-offs being common in deep learning, I think this problem is a bit understudied. For instance, object features from objects that differ a lot in distance should be easily discriminated against. Would larger $K$ help with a larger $\epsilon$? Or is this design related to hard negative sampling?
>
> **A2.** Our quasi-isometric loss addresses the negative transfer problem in multi-task learning due to two primary reasons.
> First, the quasi-isometric loss inherently has a capable margin, denoted as $(K, B)$, which ensures the distinctiveness of other sub-task classifiers.
> To illustrate, consider two objects: one from the car class and the other from the pedestrian class.
> Even though both objects are at the same depth, our loss ensures their distinguishability.
> In contrast, other contrastive losses relying on GT depth tend to consistently aggregate feature points based solely on depth, neglecting the discriminability required for other sub-tasks.
> This is further described in `R-Table 1`.
>
> Second, our quasi-isometric loss benefits from the incorporation of a local distance-preserving condition.
> This ensures a structured arrangement of the feature manifold while maintaining its intricate overall shape.
> For instance, let's imagine the feature space being modeled by a subset of the circle manifold, as depicted in `R-Figure 2-(a)`.
> From a depth perspective, this manifold represents a structured feature space since the distance of all object feature pairs along the geodesic corresponds closely with depth distance.
> However, without the local distance-preserving condition, as shown in `R-Figure 2-(b)`, the quasi-isometric loss might erroneously infer that features $p4$ and $p5$ violate property norms.
>
> On the other hand, integrating the local distance-preserving condition denoted by $\epsilon$ (`R-Figure 2-(c)`) refines the quasi-isometric loss to only consider neighbor samples.
> This approach enables a more nuanced arrangement of the feature manifold while preserving the overall shape and non-linearity of the original feature space.
>
> In summary, the pre-defined hyperparameters $(K,B,\epsilon)$ play a crucial role in determining the strictness of the local quasi-isometric property-violated condition.
> As you rightly pointed out, increasing the value of $K$ can potentially relax the strict condition set by a larger $\epsilon$.
> However, it is essential to underscore that an excessively large $K$ cannot effectively promote depth discriminative features in objects.
> Moreover, an excessively large $\epsilon$ also harms the non-linearity of the feature manifold.
>
> For empirical clarity, we conducted an experiment with MonoCon [6] + $\cal{L}_{qi}$, setting $K=5.0$ and $\epsilon=\infty$, which we deliberately set to excessively large values.
> This resulted in a degraded performance, registering a car class mAP of $17.51$ (moderate), in comparison to the $17.84$ (moderate) attained by the standard MonoCon.
> ___
> >**Q3.** Figure 2 seems interesting. However, I assume the pairs are also defined by the given $K, B,$ and $\epsilon$, which makes it less convincing.
>
> **A3.** We agree that the pre-defined hyper-parameters $(K, B, \epsilon)$ diminish the elegance of the proposed quasi-isometric loss. In the near future, we plan to consider solutions like automated parameter searches to address the issue.
> ___
> >**Q4.** What is “Ours” in supplementary Table 3? Only $\cal{L_{qi}}$ or both $\cal{L_{qi}}$ and $\cal{L_{obj}}$? I assume when ablating $K, B,$ and $\epsilon$, you should not involve $\cal{L_{obj}}$ in the comparison?
>
> **A4.**
> We appreciate your attention to detail. In `S-Table 3`, "Ours" includes both $\cal{L_{qi}}$ and $\cal{L_{obj}}$. We revisited the experiments for `S-Table 3` and updated `R-Table 2` accordingly.
> Due to space constraints, we provided only partial hyperparameter setups but will include more details on additional hyperparameter setups in the revised version.
> Notably, the trends observed in the experimental results remain consistent.
> We identified a degradation in performance with excessively large values of $K$ or $\epsilon$.
> ___
> >**Q5.** Since there is an auxiliary head for depth estimation, is the estimated depth more accurate with the quasi-isometric loss? It would be good to compare the trade-off between depth estimation and “other sub-tasks” when $\epsilon$ changes.
>
> **A5.** In `M-Table 4`, the ablation study for $\cal{L_{obj}}$ and $\cal{L_{qi}}$ shows that the marginal performance gains closely match the cumulative ones, highlighting improved depth estimation. In response to your feedback, we have conducted additional experiments adjusting epsilon and detailed the findings in `R-Table 5` to examine its impact on depth estimation relative to other tasks.
> ___

---

> > ### Comment · Reviewer_nNTt · 2023-08-13
> > **Final Rating**
> >
> > Thank the authors for preparing the additional experimental results. They addressed my concerns and I believe the results and discussions would a good addition to the paper. I think this is a strong submission and recommend "7: Accept".

---

### Official Review · Reviewer_6BaQ · 2023-07-08

**Soundness:** 3 good
**Presentation:** 3 good
**Contribution:** 3 good
**Rating:** 7
**Confidence:** 4

**Summary:**

The paper proposed a new approach to monocular 3D object detection. The critical contribution of the work is the application of metric learning to improve depth estimation and an additional head with auxiliary depth prediction. The resulting approach improves 3D object detection accuracy without increasing inference time, model size, or additional data.

The proposed method utilizes a metric learning scheme that preserves the geodesic distance between depth information and the feature space. This scheme encourages the model to extract depth-discriminative features without negatively impacting other non-depth tasks (object size, type and etc.). An auxiliary head is also introduced to enhance depth estimation, adapting from [20]. This auxiliary head improves the quality of depth estimation without impacting inference time, ensuring efficient performance.

The experimental evaluations conducted on the KITTI and Waymo datasets demonstrate the effectiveness of the proposed method. The results consistently show solid improvements in performance across various monocular 3D object detection methods.

------ Updated my score after rebuttal.

**Strengths:**

- The paper is well-written and easy to follow. The authors did a great job describing the quasi-isometric concept, its mathematics, and how the proposed loss enforces them.
- The paper is well-intuitive. The different asks in 3D object detection can create conflict in feature extraction, negatively impacting each task. The idea of using metric learning to provide a feature manyfold that balances the discrimination of depth and local structure for other tasks is promising.
- The related work section is comprehensive, covering different related areas.
- The experiment demonstrates the strength of the proposed method and generalizability on different baselines and datasets. Ablation analysis is done on the two key components to show their effectiveness and contribution.

**Weaknesses:**

- The paper provides two essential components contributing to the final improvement. But I don't think the second auxiliary head contains a lot of novelty compared to related works such as MonoCon. This limits the novelty of the paper to some extent.
- It's great that the paper compared to SupCR in ablation analysis. However, there is a missing discussion from a more theoretical standpoint on why the proposed loss is better than contrastive loss. In the end, both of the losses seem to be able to maintain a certain local distance for the other tasks.
- Part of the parameter-sensitive analysis should be moved to the main paper.

**Questions:**

- Can I ask why not use nuScenes, which seems to be more popular in monocular 3D detection in the past few years?
- How does P correspond to the feature map h in section 3.2? Is P the feature space that the map predicts? Or is a different one going through some MLP?
- Is there any systematic way for parameter choosing? Even just a starting set?
- Have the authors tried other metric learning methods that didn't seem to work?

**Limitations:**

The parameter of the loss seems to be quite sensitive to the final performance. Would it be helpful to provide guidelines or an algorithm for parameter initialization?

---

> ### Author Rebuttal · Authors · 2023-08-09
>
> >**Q1.** I don't think the second auxiliary head contains a lot of novelty compared to related works such as MonoCon. This limits the novelty of the paper to some extent.
>
> **A1.** We acknowledge your concern regarding the novelty of the auxiliary head.
> While it may not present a significant technical novelty, it uses an object-wise depth map as a subordinate task to mitigate errors in a depth task stemming from center shifting.
> This is because CenterNet-based models rely on the exact center point pixel, assuming objects as points.
> ___
> >**Q2.** There is a missing discussion from a more theoretical standpoint on why the proposed loss is better than contrastive loss. In the end, both of the losses seem to be able to maintain a certain local distance for the other tasks.
>
> **A2.** Our quasi-isometric loss addresses the negative transfer problem in multi-task learning due to two primary reasons.
> First, the quasi-isometric loss inherently has a capable margin, denoted as $(K, B)$, which ensures the distinctiveness of other sub-task classifiers.
> To illustrate, consider two objects: one from the car class and the other from the pedestrian class.
> Even though both objects are at the same depth, our loss ensures their distinguishability.
> In contrast, other contrastive losses relying on GT depth tend to consistently aggregate feature points based solely on depth, neglecting the discriminability required for other sub-tasks.
> This is further described in `R-Table 1`.
>
> Second, our quasi-isometric loss benefits from the incorporation of a local distance-preserving condition.
> This ensures a structured arrangement of the feature manifold while maintaining its intricate overall shape.
> For instance, let's imagine the feature space being modeled by a subset of the circle manifold, as depicted in `R-Figure 2-(a)`.
> From a depth perspective, this manifold represents a structured feature space since the distance of all object feature pairs along the geodesic corresponds closely with depth distance.
> However, without the local distance-preserving condition, as shown in `R-Figure 2-(b)`, the quasi-isometric loss might erroneously infer that features $p4$ and $p5$ violate property norms.
>
> On the other hand, integrating the local distance-preserving condition denoted by $\epsilon$ (`R-Figure 2-(c)`) refines the quasi-isometric loss to only consider neighbor samples.
> This approach enables a more nuanced arrangement of the feature manifold while preserving the overall shape and non-linearity of the original feature space.
> ___
> >**Q3.** Parameter-sensitive analysis should be moved to the main paper.
>
> **A3.** Thank you for your constructive feedback. We will report the parameter-sensitive analysis in the revised version.
> ___
> >**Q4.** Can I ask why not use nuScenes, which seems to be more popular in monocular 3D detection in the past few years?
>
> **A4.**
> Many monocular 3D object detection studies have favored evaluation on the KITTI [7] and Waymo [8], with fewer focusing on the nuScenes [4].
> Heeding your suggestion, we have now expanded our evaluation to include the nuScenes dataset, as presented in `R-Table 4`.
> The proposed method achieved a $2.8$% improvement in MAE, and a notable $12.3$% enhancement in Car AP, resulting in an overall performance boost of $7.6$%.
> These consistent gains across all datasets underline the robustness and generalization ability.
> ___
> >**Q5.** How does P correspond to the feature map h? Is P the feature space that the map predicts? Or is a different one going through some MLP?
>
> **A5.**
> We directly extract the object feature $\rho$ from the feature map $\mathbf{h}$ by referencing the spatial coordinate of GT coarse projected 3d center $(u,v)$.
> ___
> >**Q6.** Is there any systematic way for parameter choosing? Even just a starting set?
>
> **A6.**
> While the systematic approach to parameter selection was not a focal point of the main paper, we did identify two critical observations that assist in this process:
> First, for datasets with a large number of objects, selecting a stricter $B$ can enhance performance.
> This is achieved by approximately preserving the pseudo-geodesic distance in the $\mathbf{P}$-space. (Refer to Section B in supplementary materials)
> Second, as mentioned in the main paper, both excessively small or large values of $\epsilon$ can hinder representation learning.
> Specifically, a small $\epsilon$ risks sampling an insufficient number of property-violated object pairs.
> On the other hand, a large $\epsilon$ can compromise the non-linearity of the feature manifold within $\mathbf{P}$-space.
>
> We suggest beginning with the parameter set of $(K=1.5, B=0.5, \epsilon=10.0)$, as our experiments have shown that this configuration consistently boosts performance across multiple baselines and datasets.
> We acknowledge that the pre-defined parameters may detract from the robustness of our method.
> However, future research could address this issue, possibly through methods like automated parameter search, among other potential solutions.
> ___
> >**Q7.** Have the authors tried other metric learning methods that didn't seem to work?
>
> **A7.**
> In our main paper, we excluded results from conventional metric learning methods like SimCLR [1] and SupCon [2] since they are primarily tailored for classification tasks.
> Instead, we chose to compare ours with SupCR [3], which is designed specifically for regression tasks.
> This is because the methods [1,2] are not directly applicable to object depth estimation as a regression task.
>
> To highlight the advantages of our approach over conventional contrastive losses, we modified two renowned contrastive learning techniques: SimCLR [1] and SupCon [2].
> SupCon is specifically tailored to enhance feature learning by harnessing the information derived from the GT depth (refer to global response for details).
>
> Due to space constraints, we kindly ask you to refer to **A2-2** of **R1 (Wa1k)** for a comprehensive analysis of the experiment.
> ___

---

> > ### Comment · Reviewer_6BaQ · 2023-08-21
> >
> > I appreciate the authors' effort in the rebuttal—the rebuttal address most of my concerns.
> > I want to increase my rating based on the explanation and the new results.

---

### Official Review · Reviewer_Wa1k · 2023-07-13

**Soundness:** 3 good
**Presentation:** 3 good
**Contribution:** 3 good
**Rating:** 6
**Confidence:** 3

**Summary:**

This paper proposes a metric learning scheme to learn depth-discriminative features for object depth prediction, which helps improve the overall task of monocular 3D object detection, without negatively impacting the performance of the other sub-tasks (e.g., object class, bounding box size) wherein. Specifically,  they employ a distance-preserving function and the proposed ($K, B, ε$)-quasi-isometric loss to arrange the feature space manifold in accordance with ground-truth object depth, while preserving the non-linearity of the natural feature manifold. They also introduce an auxiliary head (in training) for object-wise depth estimation to enhance the depth quality. Experiments on datasets KITTI and Waymo show that the proposed method can be incorporated into several 3D object detection backbones for improvement.

**Strengths:**

- The proposed $(K, B, ε)$-quasi-isometric loss is mathematically explained to help learn depth-discriminative features.
- Experiments on KITTI and Waymo show the effectiveness of the proposed method, and the ablation studies are well-designed for verification including several backbones and comparison with other SOTA (e.g., SupCR).
- The paper is well written, including the problem definition, the purpose, and the background. The equations and the detailed notations help to understand the method.

**Weaknesses:**

See **Questions** below.

**Questions:**

- Are those hyper-parameters $K$, $B$ and $\epsilon$ easy to find in pratice? Will different backbone architectures require different setups of $K$, $B$, and $\epsilon$?
- According to Eq 6, the proposed Quasi-isometric loss is implemented as the contrastive loss.
   1) Are there any restrictions on the absolute number and the relative ratio of positive/negative samples/anchors?
   2) Is there any direct comparison between the proposed Quasi-isometric loss with the naive contrastive loss for metric learning in the task of monocular 3D object detection?
- It would be helpful to show some visualization results of the learned features.

**Limitations:**

- N/A for the limitations.
- I suggest the authors provide the failure cases (and the corresponding explanations) of the proposed losses on monocular 3D object detection.

---

> ### Author Rebuttal · Authors · 2023-08-09
>
> >**Q1.** Are those hyper-parameters $K, B,$ and $\epsilon$ easy to find in practice? Will different backbone architectures require different setups of $K, B,$ and $\epsilon$?
>
> **A1.** Finding the *"Optimal"* hyper-parameters can be challenging across diverse backbones and datasets. However, our experiments spanning multiple datasets and baselines consistently used the hyperparameters: $(K=1.5, B=0.5, \epsilon=10.0)$. These settings consistently enhanced the 3D object detection performance across various setups.
> ___
> >**Q2.** According to Eq 6, the proposed Quasi-isometric loss is implemented as the contrastive loss.
> - Are there any restrictions on the absolute number and the relative ratio of positive/negative samples/anchors?
> - Is there any direct comparison between the proposed Quasi-isometric loss with the naive contrastive loss for metric learning in the task of monocular 3D object detection?
>
> **A2-1.** Our quasi-isometric loss does not have rigid constraints. Most 3D detection networks, like MonoCon [6], cap at 30 objects per image. If there are no positive/negative samples, the loss returns zero, courtesy of the NT-Xent loss design.
>
> **A2-2.** Since object depth estimation is a regression-oriented task, most conventional contrastive learning methods aren't directly applicable, with the exception of SupCR [3]. Upon reviewers' requests, we adapted the two prominent contrastive learning schemes, SimCLR [1] and SupCon [2], which were originally designed for classification tasks, to enable feature learning. SupCon is specifically tailored to enhance feature learning by harnessing the information derived from the GT depth (refer to global response for details).
>
> Same as the proposed loss $\cal{L_{qi}}$, all contrastive losses were applied to object feature $\rho$.
> Our results in `R-Table 1` highlight the comparison of the proposed $\cal{L_{qi}}$ to the modified contrastive losses.
> A detailed description of each contrastive loss implementation can be found in the global response.
>
> Regarding $\cal{L_\text{SimCLR}}$, it underperforms the baseline substantially. This is because $\cal{L_\text{SimCLR}}$ focuses on extracting discriminative features of the object even when the object features share identical GT depth. When employing $\cal{L_\text{SupCon}}$ or $\cal{L_\text{SupCR}}$ as the loss function, there is a marked improvement in performance over $\cal{L_\text{SimCLR}}$.
> This underscores the advantage of integrating depth label information during the training phase.
> It is noteworthy that $\cal{L_\text{SimCLR}}$ is self-supervised and does not leverage depth label information.
>
> We experimented with SupCon's approach denoted as $\cal{L_\text{SupCon v2}}$, where only $\cal{L_\text{SupCon}}$ was used to train the backbone feature space in the early stage. Later stages saw the training of task classifiers with $\cal{L_\text{baseline}}$ while freezing the backbone. But this setup fails to predict the 3D bounding box because the other sub-task classifiers of the model could not differentiate the pre-set feature space that was defined solely by depth.
>
> In conclusion, the method incorporating our quasi-isometric loss $\cal{L_{qi}}$ significantly surpasses those utilizing other contrastive losses, demonstrating an impressive margin of $+7.6$%p to $+23.6$%p.
> This is attributed to our quasi-isometric loss, which prioritizes neighboring samples and fine-tunes the feature manifold, all the while conserving its original shape and the intrinsic non-linearity derived from various tasks.
> For further insights, kindly refer to **A2** in response to **R2 (6BaQ)**.
> ___
> >**Q3.** It would be helpful to show some visualization results of the learned features.
>
> **A3.** Following your suggestion, we included the visualization results of the feature spaces learned by our proposed method, alongside those derived from various metric learning methods in `R-Figure 1`. For clarity, the points represent projected object feature points, and their associated colors indicate depth GT values.
> ___

---

### Author Rebuttal · Authors · 2023-08-09

We thank all reviewers for their constructive comments.
For your convenience, please download and assess the **attached PDF**.
To simplify cross-referencing, figures, and tables in the main paper, supplementary materials, and rebuttal paper are denoted as `M-[Table X/Figure X]`, `S-[Table X/Figure X]`, and `R-[Table X/Figure X]`, respectively.

Below, we detail additional experiments and visualizations conducted per reviewers' requests:
___
>In response to **R1 (Wa1k)**, we included visualization results of the MonoCon [6] feature space learned by both our proposed quasi-isometric loss method and other metric learning methods using PCA in `R-Figure 1`. The points and their corresponding colors represent the projected object feature points and ground-truth (GT) depths, respectively.
___
>Addressing the inquiries from **R2 (6BaQ)** and **R3 (nNTt)**, we introduced `R-Figure 2` to explain the theoretical standpoint of the proposed quasi-isometric loss.
In this figure, $z/p$ and $\Delta \text{z}/\Delta \text{p}$ represent the depth/object-feature and Minkowski distance of the depth/object-feature pair, respectively.
Specifically, $\Delta \text{z} = d_1(z_1, z_i), \Delta \text{p} = d_2(p_1, p_i)$, with $i = \{2,3,4,5\}$ and $d_1(\cdot, \cdot), d_2(\cdot, \cdot)$ as the Minkowski distance metrics.
In `R-Figure 2-(b)` and `R-Figure 2-(c)`, the white areas represent property-satisfied zones where properties are met, while the gray areas indicate ignore zones where representation learning between object pairs is hindered due to the constraints of the $\epsilon$-ball.
___
>Based on the feedback from **R1 (Wa1k)**, **R2 (6BaQ)** and **R3 (nNTt)**, we added `R-Table 1` to compare the proposed quasi-isometric loss with other previous contrastive losses.
Since existing contrastive losses, excluding SupCR [3], are designed for classification, we adjusted each contrastive loss for object depth estimation, which is a regression problem, and a sub-task of monocular 3d object detection:
>>$\cal{L_{\text{SimCLR}}}$ [1]: We select all possible object feature pairs as negative pairs, excluding identical entity object features.
>
>>$\cal{L_{\text{SupCon}}}$ [2]: We segmented the continuous depth interval into 5-meter bins, treating them as classes, and then trained the backbone and classifier with the original SupCon loss.
>
>>$\cal{L_{\text{SupCon v2}}}$: We initially trained the backbone feature space with only $\cal{L_\text{SupCon}}$, and subsequently trained the task classifiers with $\cal{L_\text{baseline}}$ loss while freezing the backbone.
>
>>$\cal{L_{\text{SupCR}}}$: We use the same experimental setup as in the main paper.
>
>For a fair comparison, we adopted NT-Xent for loss form and cosine similarity as the metric.
All existing contrastive learning methods make the positive pairs via two-view augmentation techniques, specifically horizontal flipping and color jittering.
___
>In response to **R3 (nNTt)**, we revised and presented the performance comparison based on hyperparameters $K, B,$ and $\epsilon$ in `R-Table 2`.
___
>Responding to **R5 (eeNQ)**, we applied our quasi-isometric loss to a BEV-based 3D object detection method, ImVoxelNet [9].
The evaluation results on the KITTI [7] **$\textit{validation}$** set can be found in `R-Table 3`.
While the implementation details for ImVoxelNet remain consistent with those mentioned in the paper, we made a modification in the batch size due to GPU memory constraints. Specifically, the batch size of ImVoxelNet was adjusted to 4, which is half of its original batch size.
___
>Per the requests from **R2 (6BaQ)** and **R5 (eeNQ)**, we performed additional experiments on the nuScenes dataset [4] with the proposed method. The results are in `R-Table 4`.
We used a split [5] comprising 28,130 train and 6,019 validation images from the front camera.
The split is convertible to the KITTI format using the script [10], facilitating comparison between MonoCon and MonoCon+Ours on the nuScenes dataset.
Metrics applied include the Mean Absolute Error (MAE) of the depth of the bounding boxes and Average Precision [4] (AP), defined by matching through thresholding the 2D center distance (d) on the ground plane, rather than using Intersection over Union (IoU).
___
>Following **R3 (nNTt)**'s suggestion, we added the performance metrics of object depth estimation with varying epsilon values and the trade-off for other sub-tasks in `R-Table 5`.
___
$\mathbf{References}$

[1] *Chen, Ting, et al. "A simple framework for contrastive learning of visual representations." ICML 2020.*

[2] *Khosla, Prannay, et al. "Supervised contrastive learning." NIPS 2020.*

[3] *Zha, Kaiwen, et al. "Supervised Contrastive Regression." arXiv (2022).*

[4] *Caesar, Holger, et al. "nuscenes: A multimodal dataset for autonomous driving." CVPR 2020.*

[5] *Shi, Xuepeng, et al. "Geometry-based distance decomposition for monocular 3d object detection." ICCV 2021.*

[6] *Liu, Xianpeng, Nan Xue, and Tianfu Wu. "Learning auxiliary monocular contexts helps monocular 3D object detection." AAAI 2022.*

[7] *Geiger, Andreas, Philip Lenz, and Raquel Urtasun. "Are we ready for autonomous driving? the kitti vision benchmark suite." CVPR 2012.*

[8] *Sun, Pei, et al. "Scalability in perception for autonomous driving: Waymo open dataset." CVPR 2020.*

[9] *Rukhovich, Danila, Anna Vorontsova, and Anton Konushin. "Imvoxelnet: Image to voxels projection for monocular and multi-view general-purpose 3d object detection." WACV 2022.*

[10] https://github.com/nutonomy/nuscenes-devkit/blob/master/python-sdk/nuscenes/scripts/export_kitti.py

---

### Decision · Program_Chairs · 2023-09-21

**Decision:**

Accept (poster)

**Comment:**

This paper proposes a metric learning scheme to learn depth-discriminative features for monocular 3D object detection. It designs a new loss consisting of a quasi-isometric loss and an object-wise depth map loss to train the model using the ground-truth object depth information. Furthermore, an auxiliary head for object-wise depth estimation is used during training. All reviewers acknowledged the novelty, effectiveness, clarity, and good performance of the proposed method and recommended the acceptance of the paper. The authors' rebuttal responds well to the reviewers' concerns; the results of the additional experiments are excellent, and several reviewers raised the final rating score. For these reasons, the paper is well worth acceptance to this conference. It is strongly recommended that the final version of the paper include the discussion and experimental results provided in the rebuttal.